**TOOLS**

# High-efficacy subcellular micropatterning of proteins using fibrinogen anchors

Joseph L. Watson[1], Samya Aich[1], Benjamí Oller-Salvia[1], Andrew A. Drabek[2,3], Stephen C. Blacklow[2,3], Jason Chin[1], and Emmanuel Derivery[1]

**Protein micropatterning allows proteins to be precisely deposited onto a substrate of choice and is now routinely used in cell biology and in vitro reconstitution. However, drawbacks of current technology are that micropatterning efficiency can be variable between proteins and that proteins may lose activity on the micropatterns. Here, we describe a general method to enable micropatterning of virtually any protein at high specificity and homogeneity while maintaining its activity. Our method is based on an anchor that micropatterns well, fibrinogen, which we functionalized to bind to common purification tags. This enhances micropatterning on various substrates, facilitates multiplexed micropatterning, and dramatically improves the on-pattern activity of fragile proteins like molecular motors. Furthermore, it enhances the micropatterning of hard-to-micropattern cells. Last, this method enables subcellular micropatterning, whereby complex micropatterns simultaneously control cell shape and the distribution of transmembrane receptors within that cell. Altogether, these results open new avenues for cell biology.**

## Introduction

Micropatterning, also known as molecular printing, is the process by which molecules are precisely deposited onto a substrate of choice with micrometer resolution. Micropatterning is now routinely used in all areas of biomedical research; for example, in DNA microarrays, which have been around for decades and rely on surfaces being printed one spot at a time (Bumgarner, 2013). Decades of photolithography technology led to the advent of parallelized techniques for the affordable and straightforward manufacturing of protein micropatterns on a variety of substrates (Braunschweig et al., 2009; Falconnet et al., 2006). This led to breakthroughs in biology, allowing researchers, for example, to print cell adhesion proteins to constrain cell shape and thus reveal the physical basis of cell polarity (Théry et al., 2006) and spindle orientation during mitosis in cultured cells (Théry et al., 2007). Concomitantly, the ability to print purified proteins onto glass brought better control in in vitro reconstitution studies, which strengthened our understanding of the dynamics of cytoskeleton contractility (Reymann et al., 2012) as well as the physiology of cytoskeletal polymers (Schaedel et al., 2019; Aumeier et al., 2016). Finally, it was recently demonstrated that electron microscopy grids could be micropatterned with cell adhesion molecules (Toro-Nahuelpan et al., 2020; Engel et al., 2019). The ability, therefore, to constrain the position and shape of cells on grids is poised to solve the decade-old problem that cells adhere much more efficiently to electron-impermeable grid bars than electron-permeable carbon mesh.

Subtractive micropatterning by photolithography offers an efficient and convenient method to generate protein micropatterns on any substrate. In this technique, the substrate is first uniformly coated with an antifouling agent such as polyethylene glycol (PEG; Falconnet et al., 2006) that blocks nonspecific protein binding. This homogenous coat is then precisely etched by photoscission using a deep UV light source and a quartz photomask (Azioune et al., 2009) to allow specific adsorption of the protein of interest. While broadly adopted, this method has the inconvenient feature that each time one wants to make a novel micropattern, a new mask has to be manufactured, adding cost and time to the scientific process. As noted early on with photobleaching-based techniques (Waldbaur et al., 2012; Bélisle et al., 2009), using a microscope instead of a mask to project the illumination micropattern has the major advantage that spatial light modulators, such as digital micromirror devices (DMDs), can be used to shape the illumination pattern at will. In addition, using a microscope de facto enables multiplexed micropatterning of multiple proteins (Bélisle et al., 2009), because the micropatterned protein can be imaged and the subsequent illumination micropattern precisely aligned with it. Multiprotein micropatterning is much more tedious to achieve with masks or

[1]Medical Research Council Laboratory of Molecular Biology, Cambridge, UK;   [2]Department of Biological Chemistry and Molecular Pharmacology, Harvard Medical School, Boston, MA;   [3]Department of Cancer Biology, Dana-Farber Cancer Institute, Boston, MA.

Correspondence to Emmanuel Derivery: derivery@mrc-lmb.cam.ac.uk;   B. Oller-Salvia's present address is Institut Químic de Sarrià, Universitat Ramon Llull, Barcelona, Spain.

Rockefeller University Press
J. Cell Biol. 2021 Vol. 220 No. 2   e202009063



protein stamps due to difficulties in alignment (Eichinger et al., 2012). Importantly, while deep UV is not compatible with glass-based microscope optics, Fink et al. (2007) found that micropatterning of PEG-coated coverslips could be achieved with regular UV light with a benzophenone photosensitizer. Later elegant work by Strale et al. (2016) integrated all these developments (microscope/DMD/photosensitizer/UV) into a pioneering technique they termed light-induced molecular adsorption (LIMAP), which they demonstrated allows convenient multiprotein micropatterning in a commercial microscope.

While micropatterning offers tremendous possibilities for biomedical research, a potential limitation of the technology is the highly variable micropatterning efficiency of different proteins. In particular, micropatterning selectivity (how much protein adsorbs to the micropatterned compared with the non-patterned region) and micropatterning homogeneity (how the adsorption density varies within the micropattern) vary greatly between proteins (see also Fig. 1 and Fig. S3). For instance, if a given cell type requires a key extracellular protein to adhere to the substrate, but that protein does not micropattern efficiently, this cell type currently cannot be used in micropatterning assays. This variability between proteins is even more problematic when micropatterning multiple proteins because it implies that the different proteins will not necessarily be micropatterned at a consistent density. For complex micropatterning experiments, such as testing the effects of opposing gradients of signaling molecules, variable and inhomogeneous adsorption efficiencies are thus a problem. Furthermore, while LIMAP-mediated multiplexed micropatterning offers unprecedented avenues for biology, it may bring additional problems. In particular, because of the sequential nature of the micropatterning process, nonspecific binding between successive micropatterns is possible, where the second protein binds to the first micropattern if it has not been completely quenched (a phenomenon we refer to as cross-adsorption in this paper). In addition, the buffers and reactive oxygen species (ROS) needed for LIMAP micropatterning may not be optimal for maintaining the activity of proteins, so while a protein might micropattern well, it might have lost its activity by the time the user starts the experiment. Another general source of activity loss when micropatterning is that the direct adsorption of a protein onto the surface may induce unfolding. All these limitations may either discourage researchers from doing such complex experiments or force them to perform long and painstaking optimizations of their micropatterning process.

Here, we describe a general method to enable micropatterning of virtually any protein at a high specificity and homogeneity. Our technology relies on a protein anchor that micropatterns well, fibrinogen, which is functionalized to recognize tags commonly used in protein purification, or that can be added to already purified proteins. We show that this quantitatively improves micropatterning of hard-to-micropattern proteins on various substrates. In particular, we demonstrate robust micropatterning of Con A, which thereby enhances the micropatterning of hard-to-micropattern cells, such as *Drosophila* cells. Furthermore, we show that our technology provides an advantage for LIMAP-mediated multiprotein micropatterning as, by design, not only does it allow for all proteins to be micropatterned with similar homogeneity and low nonspecific binding, it also shields proteins of interest from any harm induced by the micropatterning process because they are bound to the fibrinogen-anchor micropatterns after said process. We also demonstrate that our technology allows the control of the subcellular localization of membrane receptors by simultaneously micropatterning cell adhesion proteins and receptor ligands at high density. Altogether, we hope that our contribution will facilitate micropatterning experiments and open new avenues for biological research.

## Results

We started by comparing the micropatterning efficiency of two proteins broadly used for micropatterning, NeutrAvidin (Aumeier et al., 2016; Schaedel et al., 2019; Portran et al., 2013) and fibrinogen (Godinho et al., 2014; Azioune et al., 2009; Carpi et al., 2011 *Preprint*). Except for Fig. 5, all micropatterns in this study were obtained following the LIMAP protocol (Strale et al., 2016) using a UV projector in a fluorescence microscope (see Fig. S1 A for design) and the photosensitizer (4-benzoylbenzyl)trimethylammonium bromide (BBTB; see Materials and methods for one-step synthesis). To quantitatively compare micropatterns throughout this paper, we evaluated both micropattern selectivity (that is, how much protein adsorbs to the micropatterned compared with the non-patterned region) and micropattern homogeneity (that is, how the adsorption density varies within the micropattern; see also Materials and methods). As seen in Fig. 1, A and B, both the specificity and the homogeneity of fibrinogen micropatterns are quantitatively higher than equivalent NeutrAvidin micropatterns on polylysine (PLL)-PEG–coated glass. As micropatterning is an adsorption process, its efficiency is expected to vary according to the buffer composition and according to changes in protein folding and surface charges. Accordingly, we found that fibrinogen micropatterning is quantitatively improved in low-salt carbonate buffer compared with PBS buffer (Fig. S1, B and C), consistent with the known propensity of fibrinogen to precipitate in the presence of salts. For this reason, all micropatterns in this paper were generated in carbonate buffer (see Materials and methods). Importantly, fibrinogen displays remarkably low binding to non-micropatterned PLL-PEG substrates (see Fig. S1, E and F; and Materials and methods): the amount of fibrinogen-ATTO488 wrongly going onto the unpatterned PLL-PEG is only 0.40 ± 0.06% (mean ± SEM; $n = 8$) that of the amount of fibrinogen-ATTO488 going to its intended micropattern (0.66 ± 0.04% for fibrinogen-Alexa647; $n = 8$). In other words, it is not that fibrinogen binds everywhere, although better on the micropatterned region, but rather that fibrinogen binds nearly exclusively to the intended micropatterned region.

The intrinsic properties of fibrinogen also facilitate multiprotein patterning. One key advantage of using the LIMAP protocol within a microscope (Strale et al., 2016) is that it enables straightforward multiprotein micropatterning, as the user can both image one micropattern and, in the same instrument,

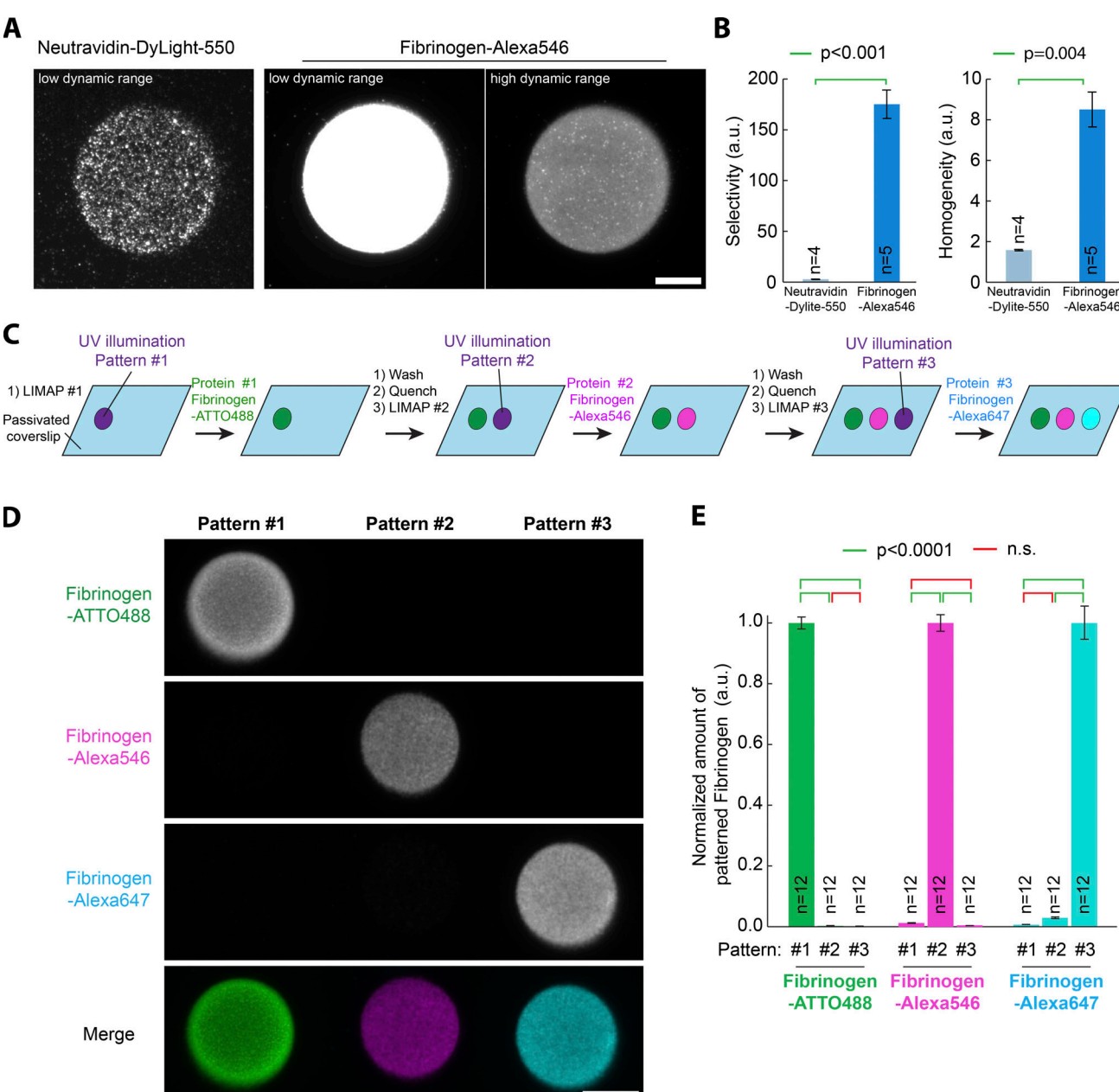

Figure 1. **High specificity and selectivity of fibrinogen micropatterning on PLL-PEG surfaces. (A)** Fibrinogen-Alexa546 (50 µg/ml) and NeutrAvidin-Dylight-550 (50 µg/ml) were micropatterned on PLL-PEG–coated glass using LIMAP with identical UV exposure and micropattern shape. After washing, red fluorescence of the micropatterns was imaged by TIRF microscopy (TIRFM) using identical settings (left and middle). Alternatively, a higher dynamic range lookup table was applied to the image on the right. **(B)** Quantification of the effects seen in A. Mean ± SEM of the selectivity and homogeneity (see Materials and methods). Statistics were performed using a Mann–Whitney rank-sum test. n, number of micropatterns measured. Fibrinogen quantitatively micropatterns better than NeutrAvidin. **(C)** Scheme illustrating the different steps for sequential multiplexed micropatterning of three fibrinogens labeled with different fluorophores (ATTO488, Alexa546, and Alexa647). **(D)** Multiplexed micropatterning of fibrinogen-ATTO488, Alexa546, and Alexa647 (50 µg/ml) using the scheme depicted in C onto PLL-PEG–coated glass. Note that there is high specificity of the fibrinogen for their specific micropattern and minimum overlap between the three fluorescent fibrinogens. **(E)** Quantification of the effects seen in D. The fluorescence of each fibrinogen was measured on the three successive micropatterns and normalized to the intensity of their respective micropattern. Mean ± SEM. Statistics were performed using a one-way ANOVA test followed by a Tukey post hoc test (P < 0.0001). n, number of micropatterns measured. Note that the amount of fibrinogen deposited onto the nonintended micropatterns is minimal. Scale bars, 10 µm. n.s., not significant.

readily align a second micropattern with respect to this micropattern (ad infinitum). However, a potential issue is cross-adsorption, where the second protein binds to the first protein pattern. This can have two origins: either the second protein binds to the first, or the first pattern was not completely saturated by the first protein (or not subsequently efficiently quenched), and thus the second protein can adhere there. Fibrinogen combines two advantages to alleviate these issues: (1) soluble fibrinogen does not bind to micropatterned fibrinogen, and (2) fibrinogen micropatterns can be efficiently quenched

with respect to fibrinogen (that is, soluble fibrinogen binds very little to a quenched fibrinogen micropattern). To highlight these properties, we thought to conduct sequential micropatterning of fibrinogen labeled with three different fluorescent dyes: ATTO488, Alexa546, and Alexa647 (Fig. 1 C). As seen in Fig. 1 D, all three fibrinogens adhere to their intended micropatterns, with minimal cross-adsorption to the other patterns. Quantification revealed that the amount of the second micropatterned protein, fibrinogen-Alexa546, wrongly going onto the fibrinogen-ATTO488 pattern is only 1.3 ± 0.05% (mean ± SEM; $n$ = 12) of the amount going to its intended pattern (see Fig. 1 E and Materials and methods). Similarly, the amount of the third micropatterned protein, fibrinogen-Alexa647, going to the wrong fibrinogen-ATTO488 or fibrinogen-Alexa546 patterns is minimal (0.7 ± 0.04% and 3.0 ± 0.3%, respectively, compared with the amount going to its intended pattern; mean ± SEM; $n$ = 12). Similar results were obtained in sequential patterning of two (not three) fibrinogens (Fig. S1, D–F).

We thus thought to exploit the inherent high micropatterning efficiency of fibrinogen to improve the micropatterning of hard-to-micropattern proteins by conjugating them together. Fibrinogen would thus act as the "anchoring" moiety, ensuring that both micropatterning selectivity and homogeneity are high and reproducible, while the protein of interest would provide the "specificity" part, bringing the function required for the experiment. This would also potentially improve accessibility, and thereby activity, of the micropatterned protein, as not all of the micropatterned protein would be facing the glass, which may occur with direct micropatterning due to potential preferred orientation of protein deposition. Constraining protein orientation would also mitigate potential surface-induced protein unfolding. Fibrinogen is an ideal anchoring moiety for this as it is commercially available in gram scales, is easy to work with, and contrary to other commonly available proteins like BSA, precipitates natively with minute amounts of ammonium sulfate, a property we could exploit to facilitate the separation of functionalized products (see Materials and methods). We thus established robust protocols for the biochemical functionalization of fibrinogen, allowing it to be conjugated to virtually any protein of interest, as well as permitting its binding to commonly used purification tags.

The conjugation method relied on the modification of fibrinogen-exposed amines with a heterobifunctional cross-linker that enabled conjugation with the target protein in a two- or three-step procedure (see Fig. S2 and Materials and methods). In particular, we derived fibrinogen-GFP, fibrinogen–Con A (a lectin that binds to insect cells; Rogers et al., 2002), fibrinogen-NeutrAvidin (Fig. S2 A) to bind to biotinylated targets, fibrinogen–GFP-binding peptide (GBP) to bind to GFP-tagged proteins (GBP is a nanobody against GFP; Fig. S2 B; Rothbauer et al., 2008) and fibrinogen-biotin (Fig. S2 C) to bind to streptavidin/NeutrAvidin fusions, as well as to biotinylated targets using a NeutrAvidin "sandwich" (fibrinogen-biotin::NeutrAvidin::biotinylated protein of interest). Importantly, since fibrinogen contains numerous exposed amines, it can be functionalized with multiple molecules. For instance, biotin and ATTO490LS can be combined (fibrinogen-biotin-ATTO490LS) to micropattern biotinylated targets while at the same time facilitating micropattern alignment by fluorescence

microscopy (see below). Note that these conjugation protocols are general and apply to virtually any protein of interest. Furthermore, these optimized protocols are straightforward, use commercially available chemicals, and do not require any specialized biochemistry equipment such as fast protein liquid chromotography (FPLC).

As a proof of concept, we compared the micropatterning efficiency of a biotinylated target, BSA-biotin-Alexa647, through either NeutrAvidin or fibrinogen-NeutrAvidin micropatterns. This is a more relevant situation than just comparing the micropatterning efficiency of fibrinogen-NeutrAvidin versus NeutrAvidin, as what matters for experiments is the density of micropatterned biotin-binding sites, which is not necessarily the same as the density of micropatterned NeutrAvidin. As it is always the same fluorescent protein being micropatterned, in addition to determining the micropattern selectivity and homogeneity, we also estimated the amount of protein being specifically bound to the micropattern from the fluorescence intensity. As seen in Fig. 2, A and B, fibrinogen-NeutrAvidin provides a fourfold improvement of the selectivity and homogeneity of micropatterned biotinylated targets, as well as the 13-fold improvement of the amount of protein being specifically micropatterned. This was true for all fibrinogens; see, for instance, fibrinogen-GFP offering higher selectivity, homogeneity, and amount of micropatterned protein than GFP alone (Fig. 2 C; see also Fig. 2 D for quantification). Importantly, while all experiments described until now were performed on PLL-PEG–coated glass, the fibrinogen toolkit described here also improves micropatterning on other commonly used substrates, such as PEG-silane (Aumeier et al., 2016; Schaedel et al., 2019; Portran et al., 2013); see Fig. S3.

We then thought to apply our technology to multiprotein patterning. As mentioned above, sequential micropatterning using LIMAP has three potential issues: (1) lack of homogeneity between micropatterns due to the high variability of micropatterning between proteins, (2) cross-adsorption between successive micropatterns because of incomplete quenching, and (3) potential activity loss due to inappropriate buffers and/or ROS generation. An additional issue is that while being able to image a micropattern in the microscope to align the next one is a major advantage of the technique, this requires the first micropatterned protein to be fluorescent. However, fluorescent derivatives of proteins are not necessarily as active as their unlabeled counterparts: for instance, we found that fluorescent NeutrAvidin (NeutrAvidin–Dylight-550), while micropatterning fine on fibrinogen-biotin, displayed a more than sixfold reduction in its ability to bind to biotinylated targets compared with unlabeled NeutrAvidin (Fig. S3). Doping the unlabeled protein to micropattern with some other fluorescent protein as a tracer is not a solution either, because, as seen in Fig. 1, micropatterning efficiencies widely differ between proteins, so the fluorescent tracer will not necessarily be a reliable proxy for the amount of unlabeled protein to be micropatterned.

The fibrinogen toolkit described here could address all these limitations at once. First, it would ensure that micropatterning homogeneity is similar for all micropatterned proteins, as it is always the same protein anchor that is micropatterned

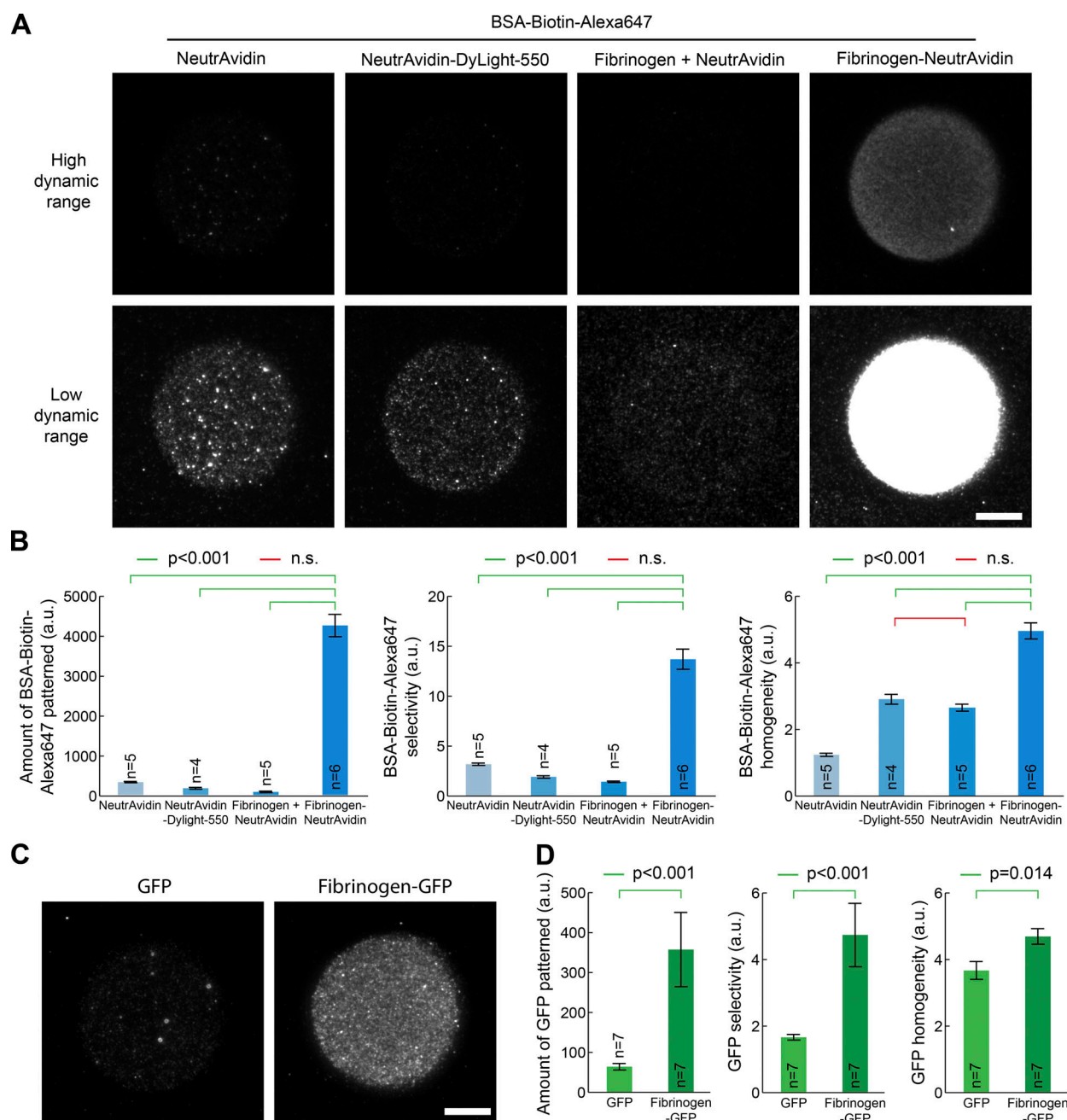

Figure 2. **Fibrinogen anchors improves selectivity and homogeneity of micropatterns on PLL-PEG surfaces. (A)** NeutrAvidin, NeutrAvidin-DyLight-550, NeutrAvidin mixed with fibrinogen, or NeutrAvidin fused to fibrinogen (Fibrinogen-NeutrAvidin) were micropatterned on PLL-PEG–coated glass using LIMAP with identical UV exposure, micropattern shape, and protein concentration (50 µg/ml). After micropattern quenching and washing, BSA-biotin-Alexa647 (5 µg/ml) was added for 5 min. The sample was then washed, and BSA-biotin-Alexa647 fluorescence was imaged by TIRFM. Two different dynamic ranges for visualization were used for each sample (top versus bottom line) so that each lane could be represented with the same dynamic range. **(B)** Quantification of the amount of protein bound to the micropattern (left), micropattern selectivity (middle), and homogeneity (right) in the sample presented in C (mean ± SEM). Statistics were performed using a one-way ANOVA test followed by a Tukey post hoc test after log10 transformation of the data (P < 0.001). Fibrinogen-NeutrAvidin enhances significantly the selectivity and the homogeneity of BSA-biotin-Alexa647 micropatterns. **(C)** GFP and fibrinogen-GFP were micropatterned on PLL-PEG–coated glass using LIMAP with identical UV exposure, micropattern shape, and protein concentration (50 µg/ml). After washing, GFP fluorescence was imaged by TIRFM. **(D)** Quantification of the amount of protein selectively bound to the micropattern (left), as well as micropattern selectivity (middle) and homogeneity (right) in the sample presented in C (mean± SEM). Statistics were performed using a Student's t test (after log10 transformation of the data for left and middle panels). The selectivity and homogeneity of fibrinogen-GFP micropatterns are significantly better than those of GFP alone. n, number of micropatterns measured. Scale bars, 10 µm. n.s., not significant.

(fibrinogen). Second, sequential micropatterning of different fibrinogen "flavors" is extremely efficient with minimal cross-adsorption (see above; Fig. 1, C–E; and Fig. S1, D–F). Third, fibrinogen anchors functionalized against common purification tags would enable one to first micropattern all the anchors, then add the proteins of interest in their buffer of choice as the last step of the sample preparation just before the actual experiment starts (Fig. 2 A), therefore ensuring that the activity of the proteins of interest is conserved. Last, each nonfluorescent fibrinogen anchor can be doped with trace amounts of fluorescent fibrinogen for micropattern alignment purposes without the risk that these trace amounts might affect the micropatterning efficiency of the fibrinogen anchor. This last issue can also be alleviated by fluorescently labeling fibrinogen anchors (Fig. 4, Fig. 6, and Fig. 7).

As a proof of concept, we micropatterned two proteins, GFP and biotin-BSA-Alexa647, using fibrinogen-GBP and fibrinogen-NeutrAvidin anchors, respectively (Fig. 3 A). As seen in Fig. 3 B, fibrinogen anchors enable reliable micropatterning of these two proteins. Cross-contamination between the two proteins is quantitatively little: the amount of GFP wrongly going onto the biotin-BSA-Alexa647 micropattern is only 5.0 ± 0.3% (mean ± SEM; $n$ = 7) of the amount going to its intended micropattern (Fig. 3 C). Similarly, the amount of biotin-BSA-Alexa647 going to the wrong GFP pattern is only 0.9 ± 0.1% ($n$ = 7) of the amount going to its intended micropattern (Fig. 3 C). In addition, micropattern homogeneity was similar for both proteins (Fig. 3 D), suggesting that both proteins were micropatterned at a consistent density. We found that multiplexed micropatterning was very reproducible with this protocol and found that the major source of variability between samples rather came from occasional accidental drying of the coverslip during the micropatterning process (Fig. S4, A and B).

As mentioned above, it is not enough that a protein micropatterns well; its activity must also be maintained on the micropattern. Loss of activity during micropatterning can have multiple origins: the micropatterning process itself can be detrimental to proteins due to ROS generation and buffer composition when doing LIMAP, but also, more generally, since micropatterning is an adsorption-based process, proteins will micropattern with their "stickier" side facing the glass, which could be the face harboring the active site, therefore affecting activity or inducing unfolding. Fibrinogen anchors alleviate the buffer/ROS issue, but they potentially could also improve orientation and conformational freedom, as the protein of interest is not itself micropatterned but rather attached to the micropattern via a flexible linker.

We thought to test this hypothesis using a commonly used assay relying on adsorbed proteins, namely the microtubule (MT) gliding assay (Fig. 4; Bachand et al., 2014). In this assay, kinesin molecular motors are adsorbed onto glass, and motors with their motor domain facing away from the glass can then move MTs around. While a biotinylated fragment comprising the motor domain and the dimerizing coiled-coil of *Drosophila* kinesin 1 (noted Kin1-biotin; Subramanian and Gelles, 2007) is efficiently micropatterned onto PLL-PEG substrates (Fig. 4, A–D), its activity is dramatically reduced when doing so: MTs move slowly (Fig. 4, C and J, for quantification; and see also Video 1) and frequently pause during their motion (Fig. 4, C and K, for quantification).

Conversely, micropatterning the same Kin1-biotin through a "sandwich" on top of fibrinogen-biotin-ATTO490LS and NeutrAvidin (Fig. 4, E–H), restored expected gliding speeds (Fig. 4, G and J, for quantification) with very rare pauses (Fig. 4, G and K, for quantification; and see also Video 1). Importantly, in all these experiments, we only detected motion in the micropatterned regions (Fig. 4 I), suggesting that MTs bound to the surface outside the micropattern were not bound by kinesin but rather by nonspecific interactions with the surface. This demonstrates that fibrinogen anchors are a reliable way to ensure high activity of micropatterned proteins, most likely by ensuring a high density of correctly oriented proteins and by mitigating surface-induced denaturation of the micropatterned proteins of interest.

We then decided to apply our fibrinogen technology to facilitate applications that would otherwise be hard with existing micropatterning techniques. Insect cells like *Drosophila* S2 cells require lectins such as Con A to adhere on glass (Rogers et al., 2002). However, Con A, in our hands, does not micropattern well (Fig. 5 A, compare "pattern" channel with fibrinogen for the small micropatterns on the right side of each image), thereby limiting the possibility of adhering these cells to small micropatterns, and therefore to micropattern single S2 cells. Note that this is mainly a problem for small micropatterns, as cells manage to adhere partially to large micropatterns (Fig. 5 A, left side of all images). We thus derived fibrinogen–Con A (Fig. S2 A), which dramatically improves micropatterning efficiency of S2 cells, even onto small, single-cell micropatterns (Fig. 5 A, compare "merge" channel between Con A and fibrinogen–Con A; see also Fig. 5, B and C, for quantification of the cell density and specificity on the small micropatterns). The improvement in adhesion was even clearer with medium-sized micropatterns, to which several cells adhered per micropattern (Fig. S4 C).

We then thought to combine all the advantages of our technology to open a new avenue in micropatterning by achieving subcellular micropatterning, whereby the position of proteins within cells can be imprinted from the outside via a micropattern. To achieve this, we made dual micropatterns with one micropattern to anchor the cell and a second micropattern to relocalize a transmembrane receptor via an active ligand presented in the right orientation (Fig. 6 A; and Fig. 7, A and D). Obviously, achieving this quantitatively relies on the ability to achieve multiplexed micropatterning of proteins while maintaining their activity and proper orientation. As a proof of concept, we used NIH/3T3 cells stably expressing a model receptor composed of a transmembrane segment fused to an extracellular GBP and an intracellular mScarlet (GBP-TM-mScarlet), and dual micropatterns composed of a fibrinogen/fibronectin anchoring micropattern, and a fibrinogen-biotin-ATTO490LS::streptavidin-GFP-GFP relocalizing micropattern. Streptavidin-GFP-GFP refers to a fusion between streptavidin and two copies of GFP. Importantly, this setup achieved specific and quantitative relocalization of the GBP-TM-mScarlet construct onto the GFP micropattern (Fig. 6, B and C). Protein relocalization happened in a matter of minutes when cells entered in contact with the GFP-positive part of the micropattern during spreading and was then stable over time (Fig. 6, D and E; and Video 2). Importantly, this result showcases

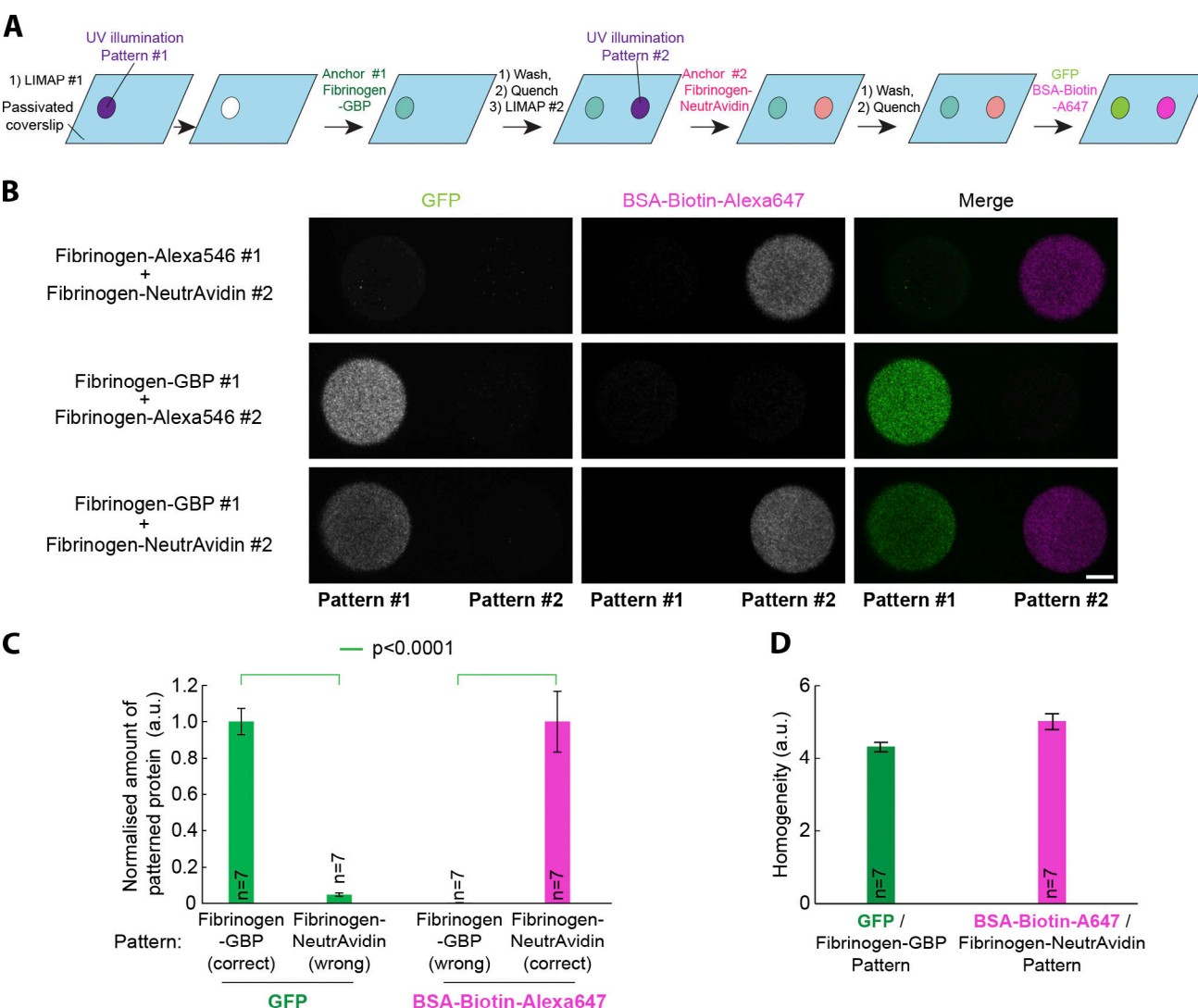

Figure 3. **Fibrinogen anchors facilitate multiplexed micropatterning of proteins. (A)** Scheme illustrating the different steps for multiplexed micropatterning using fibrinogen anchors. Note that the two proteins to be micropatterned (GFP and BSA-biotin-Alexa647) are added together after the micropatterning process. This could help maintain protein activity, as proteins can be added in their optimal buffer (rather than the optimal micropatterning buffer) and are not exposed to UV-induced ROS or the BBTB. **(B)** Bottom line: Multiplexed micropatterning of GFP and BSA-biotin-Alexa647 (5 µg/ml) with fibrinogen-GBP and fibrinogen-NeutrAvidin anchors (50 µg/ml) using the scheme depicted in A onto PLL-PEG–coated glass. Top and middle lines: Controls of bottom panel by exchanging fibrinogen-GBP with fibrinogen-Alexa546 (top line) or by exchanging fibrinogen-NeutrAvidin with fibrinogen-Alexa546 (middle line) followed by coinjection of GFP and BSA-biotin-Alexa647. Note that there is high specificity of the proteins of interest for their respective anchor and minimum overlap between the two proteins. This implies that the successive micropatterns of fibrinogen anchors have been efficiently quenched (otherwise the proteins of interest could bind to the wrong micropattern). **(C)** Quantification of the increased specificity of GFP and BSA-biotin-Alexa647 for their respective fibrinogen-GBP or fibrinogen-NeutrAvidin anchor, expressed as the amount protein going to the wrong patterns as a fraction of the amount going to the right pattern (mean ± SEM). Statistics were performed using an unpaired t test. **(D)** Similarity of the micropatterning homogeneity of GFP on fibrinogen-GBP micropatterns and BSA-biotin-Alexa647 on fibrinogen-NeutrAvidin micropatterns (mean ± SEM). n, number of micropatterns measured. Scale bar, 10 µm.

the advantages of the high micropatterning densities and modularity that are achievable with our fibrinogen toolbox, as micropatterns offering lower GFP densities, such as direct micropatterning of GFP or fibrinogen-GFP with a low degree of labeling, did not induce noticeable GBP-TM-mScarlet relocalization (Fig. S5, A and B).

We then generalized this concept to endogenous receptor/ligand pairs to establish controlled signaling platforms at the surface of cells. First, a micropattern of a soluble ligand, namely EGF, proved efficient at relocalizing its cognate receptor (EGFR) when anchored via a fibrinogen-biotin::NeutrAvidin sandwich (Fig. 7, A and B; see

also Fig. 7 C for quantification). This was not observed without the addition of biotinylated EGF (Fig. 7, B and C). Importantly, the poor direct micropatterning of fluorescent EGF (Fig. S5 C) made this dual-patterning assay impossible without fibrinogen anchoring. The modular nature of the fibrinogen-anchoring technology also means that even higher densities of active EGF could presumably be achieved using a ligand amplification strategy, akin to that used to increase GBP-TM-mScarlet relocalization (Fig. 6, A and B). Interestingly, with EGFR relocalization, we found that clathrin was also relocalized to the EGF::EGFR micropattern (Fig. 7 B; see

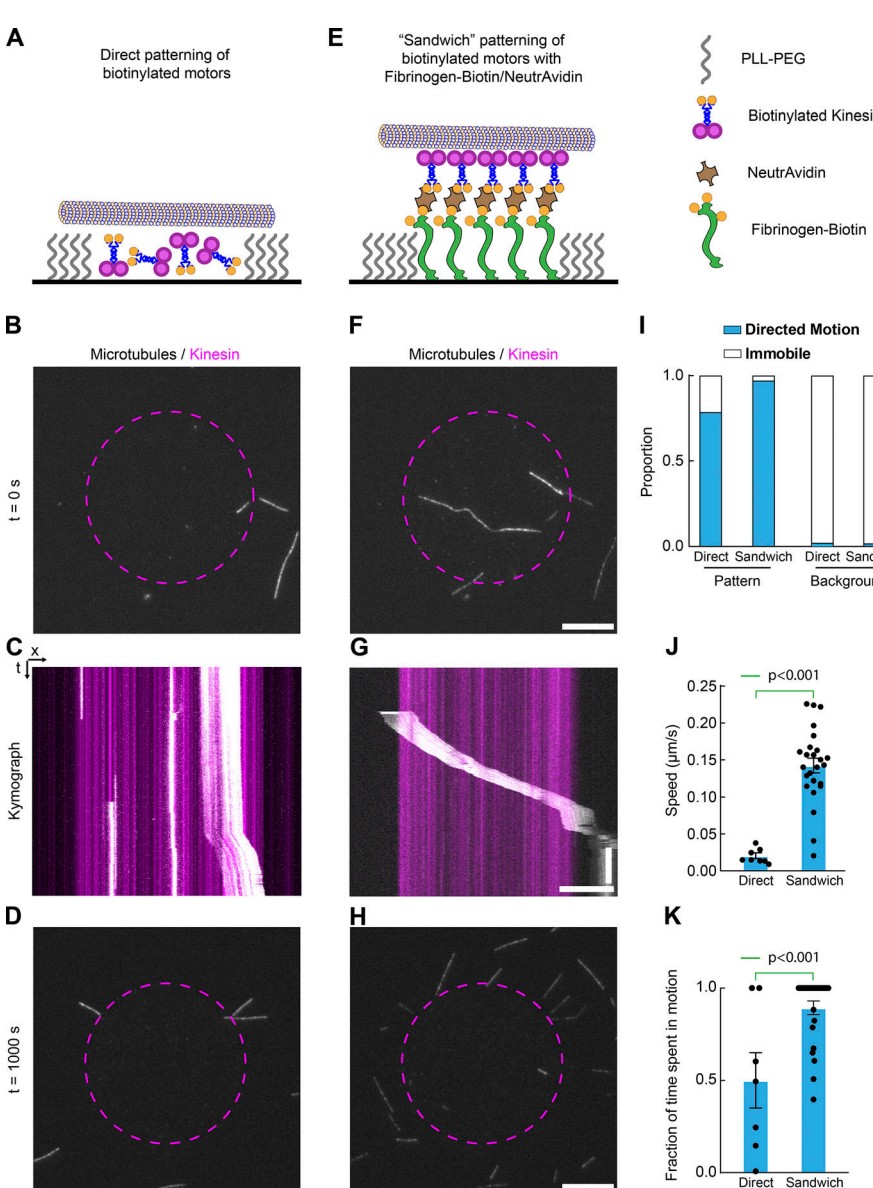

Figure 4. **Fibrinogen anchors facilitate micropatterning of active motors. (A–H)** Biotinylated-Kinesin1 motors (Kin1-Biotin) were micropatterned on PLL-PEG–coated glass using LIMAP either directly (A–D) or indirectly through a fibrinogen-biotin-ATTO490LS::NeutrAvidin sandwich (E–H; see Materials and methods). After washing and quenching, GMPCPP-stabilized fluorescent microtubules (MTs) were added in the presence of ATP and their motion observed by TIRFM. Dashed purple lines delineate the kinesin micropattern as imaged either through post-labeling of Kin1-Biotin by streptavidin-Alexa647 for direct micropatterning (A–D) or by fibrinogen-biotin-ATTO490LS fluorescence for the indirect micropatterning (E–H). **(I)** Quantification of the proportion of motile MTs in all conditions reveals that MTs outside the micropattern are immobile, in contrast to MTs landing inside the micropattern (n = 14/34/47/112, respectively). **(J and K)** Quantification of the speed (J; mean ± SEM) and fraction of time spent moving processively (K; mean ± SEM) reveals that MTs move faster and with fewer pauses on fibrinogen-biotin–mediated Kin1-Biotin micropatterns (directly micropatterned, n = 7; sandwich, n = 25). This suggests that indirect micropatterning of Kin1-Biotin through fibrinogen-biotin ensures high activity of the motor on the micropattern. Statistics in J and K were performed using unpaired t tests. n, number of MTs. Scale bars, 10 μm (B, D, F, and H) and 10 μm/1 min (C and G). t, time.

also Fig. 7 C for quantification), suggesting that the EGF micropattern traps EGFR in an intermediate state along the endocytic pathway, as expected if the micropattern activates the physiological signaling cascade downstream of EGFR activation but prevents EGFR endocytosis because of the strong link to the micropattern. Similarly, a micropattern of a ligand normally exposed at the surface of cells, Delta, proved efficient at relocalizing its cognate receptor, GFP-Notch, in live cells (Fig. 7, D–F). As before with the synthetic GFP::GBP interaction, relocalization of GFP-Notch by micropatterned Delta occurred in the time scale of minutes as cells spread onto the micropattern and remained stable over time (Fig. S5, D and E; and Video 3).

## Discussion

The technology developed in this paper allows the micropatterning of virtually any protein of interest with high selectivity and homogeneity. We envision that this will open new avenues for cell biology

applications, for instance, to micropattern cell types that could not be well micropatterned due to the low micropatterning efficiency of their extracellular matrix of choice. In particular, the enhanced micropatterning of *Drosophila* cells that we achieved in this work (Fig. 5) opens a new avenue for cell biology, as it will allow us, through the isolation of primary *Drosophila* cells, to combine quantitative, geometrically defined cell culture with the genetic tractability of *Drosophila*. Importantly, while we focused here on UV micropatterning in a microscope (Strale et al., 2016), our technology is general and applies to any light-induced micropatterning technique, such as Deep UV-Quartz-mask–based techniques (Fig. 5; Azioune et al., 2009). The ability to functionalize fibrinogen with nanobodies could also be a benefit for the clustering of receptors against which antibodies have been developed due to the elegant work by Pleiner et al. (2018), who recently released an extensive toolkit of nanobodies targeting nearly all commonly used primary antibodies. Indeed, these nanobodies are compatible with their rational cysteine-based nanobody engineering protocol (Pleiner et al.,

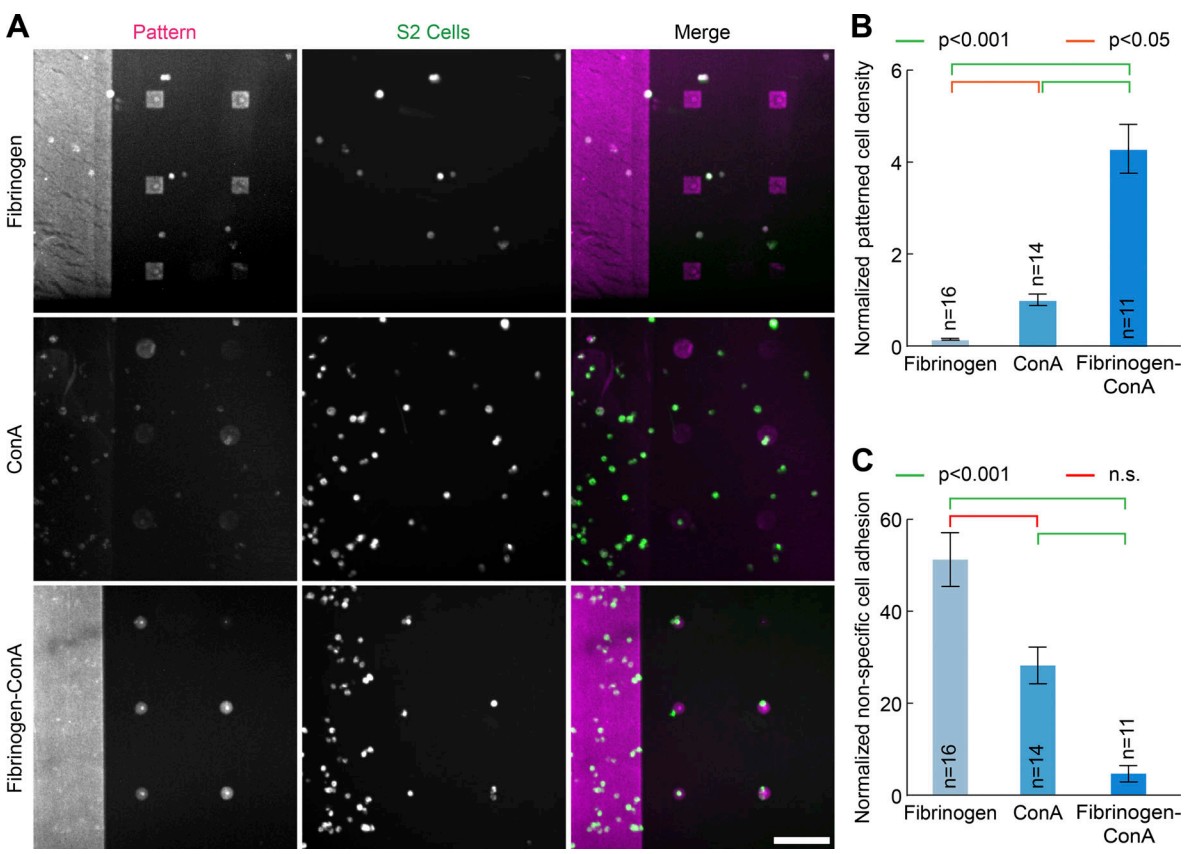

Figure 5. **Fibrinogen anchors facilitates the micropatterning of hard-to-pattern cells. (A)** Fibrinogen (doped with 10% fibrinogen-Alexa546), Con A (doped with 10% rhodamine–Con A), or fibrinogen–Con A (doped with 10% fibrinogen-Alexa546) were micropatterned at 50 µg/ml onto PLL-PEG–coated glass using deep-UV and a chromium mask. Coverslips were washed, and S2 cells were added for 1 h before addition of SiR-tubulin to label cells for 30 min. After washing, cells and micropatterns were imaged by spinning-disc confocal microscopy. While S2 cells do manage to adhere to larger Con A micropatterns, micropatterning efficiency of Con A is lower than that of fibrinogen (compare left panels), and therefore fibrinogen–Con A enables micropatterning of S2 cells onto small, single-cell micropatterns. Note that fluorescence in the "pattern" channel cannot be directly compared as Con A and fibrinogen are not functionalized with the same fluorophores. **(B and C)** Quantification of the effects seen in A (mean ± SEM; fibrinogen, *n* = 16; Con A, *n* = 14; fibrinogen–Con A, *n* = 11). Normalized micropatterned cell density (B) is significantly higher on small fibrinogen–Con A micropatterns compared with Con A or fibrinogen micropatterns, while the normalized nonspecific adhesion (C) is higher for fibrinogen and Con A than fibrinogen–Con A (see Materials and methods for details). Statistics in B were performed using a one-way ANOVA test followed by a Tukey post hoc test (P < 0.001), while statistics in C were performed using a Kruskal–Wallis test. n, number of fields of view analyzed. Scale bar, 100 µm. n.s., not significant.

2015), which we rely on for fibrinogen functionalization with GBP. Importantly, while we focused here on a few commonly used protein tags, there is no reason why other binders/ligands could not be fused to fibrinogen, such as benzylguanine (to bind to SNAP-tagged proteins), Strep-Tactin (to bind to Strep-Tag–tagged proteins; Schmidt and Skerra, 2007), amylose (to bind to maltose-binding protein–tagged proteins), or Spycatcher/Snoopcatcher (to bind to SpyTag/SnoopTag-tagged proteins; Veggiani et al., 2016).

An additional major advantage of our method is that it ensures that micropatterned proteins maintain their activity, which was best exemplified by our successful micropatterning of active molecular motors (Fig. 4). This is especially important for sequential micropatterning using LIMAP, as the ROS generated by UV light to micropattern the second protein can potentially be harmful to the first micropatterned protein. This improvement in activity is achieved by first micropatterning fibrinogen anchors that specifically bind to purification tags, followed by addition of the tagged proteins of interest. As the proteins of interest are added after the actual micropatterning

process, they can be added in the optimal buffer for their stability/activity. Conversely, the fibrinogen anchor is micropatterned in its optimal buffer for micropatterning efficiency, thereby removing the need to compromise between the two. It must be emphasized that our technology also maximizes activity by mitigating preferential orientation effects, as it ensures that binding to the micropattern occurs at the level of the protein tag, usually added to flexible N- or C-terminal regions, rather than on the "stickiest" face of the protein, which could also happen to be where the protein active site is. In particular, this allowed us to micropattern molecular motors specifically via their C-terminal tail away from their N-terminal motor domain, thereby maintaining their activity (Fig. 4). We envision that these key assets will provide major advantages for in vivo studies, to micropattern active ligands in the right orientation to bind to membrane receptors, but also for in vitro studies, for multiplexed micropatterning of various molecular motors to pave the way toward the in vitro reconstitution of complex cytoskeleton landscapes, akin to those found in cells.

Figure 6. **Fibrinogen anchors enables the subcellular micropatterning of a synthetic receptor fused to a cortical protein of interest. (A)** Experimental scheme: NIH/3T3 cells stably expressing GBP-TM-mScarlet were allowed to spread on dual micropatterns of fibronectin/fibrinogen-Alexa647 and fibrinogen-biotin-ATTO490LS::streptavidin-GFP-GFP, and were then imaged live by TIRF microscopy. **(B)** Efficient relocalization of the GBP-TM-mScarlet construct onto an area defined by the extracellular GFP micropattern in live cells. **(C)** Quantification of the effects seen in B (mean ± SEM). Statistics were performed using a Student's *t* test. n, number of cells analyzed. **(D)** Cells as in B were imaged by TIRF microscopy during spreading to evaluate the kinetics of GBP-TM-mScarlet recruitment onto the GFP micropattern. Fibrinogen and GFP micropatterns are outlined in blue and green dashed lines, respectively. **(E)** quantification of the effects seen in D (mean ± SEM; number of cells analyzed: 11 for control and 32 for Streptavidin-GFP-GFP). Scale bar, 10 µm. Ctrl, control; Strept., streptavidin.

In addition to ensuring protein activity, our method also minimizes cross-adsorption between successive micropatterns as it simplifies micropattern quenching. In other words, not only are the proteins active, but they are also at the right place (Fig. 1, Fig. 3, and Fig. S1). Indeed, because it is always virtually the same protein that is micropatterned, this simplifies the determination of the quenching conditions for each specific experiment. Coupled to the constant homogeneity between micropatterns

brought by our method, we envision that this property will be a key advantage for multiprotein applications where control over the relative micropatterning density between proteins is important, for example, in testing the cellular response to multiple gradients of signaling molecules.

While not a concern for in vitro experiments, fibrinogen might not always be the anchor of choice when working with cells, as it is itself a bioactive molecule. First, obviously,

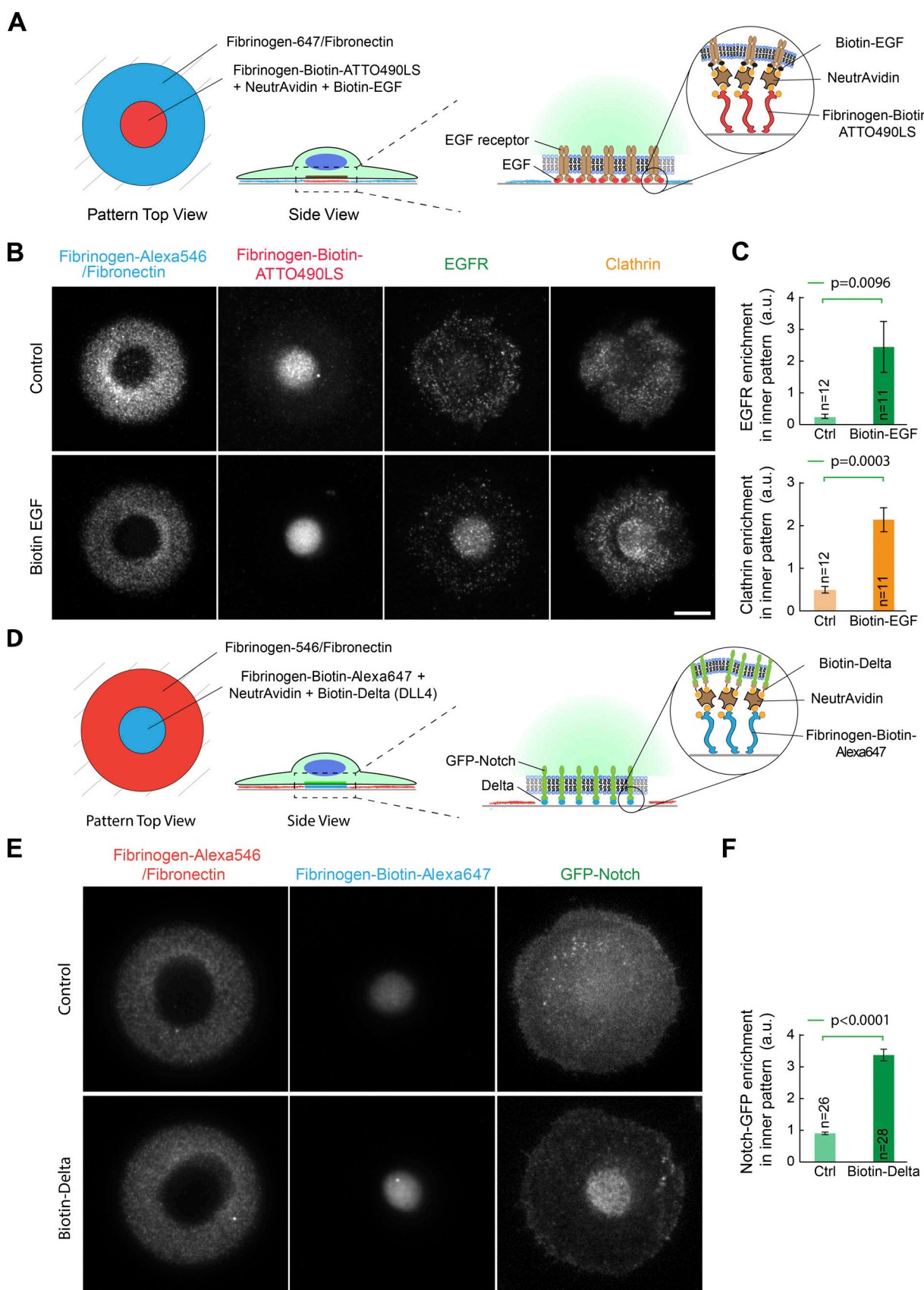

Figure 7. **Fibrinogen anchors enables the subcellular micropatterning of endogenous receptors. (A)** Experimental scheme. Serum-starved HeLa cells were allowed to spread on dual micropatterns of fibronectin/fibrinogen-Alexa546 and fibrinogen-biotin-ATTO490LS::NeutrAvidin/biotin-EGF, then fixed and

processed for immunofluorescence against EGFR and clathrin. **(B)** Efficient relocalization of the EGFR onto the active EGF micropattern. Note that clathrin is also recruited onto the micropattern. **(C)** Quantification of the effects seen in D (mean ± SEM). **(D)** Experimental scheme. U2OS cells stably expressing GFP-Notch1 were allowed to spread on dual micropatterns of fibronectin/fibrinogen-Alexa647 and fibrinogen-biotin-ATTO490LS::NeutrAvidin::biotin-DLL4 and imaged live 20 min later. **(E)** Efficient relocalization of GFP-Notch1 onto the active DLL4 micropattern. **(F)** Quantification of the effects seen in E (mean ± SEM). All statistics in this figure were performed using Student's *t* tests. n, number of cells analyzed. Scale bar, 10 µm. Ctrl, control.

fibrinogen anchors are probably not suited to study cell types involved in the biology of fibrinogen, like platelets. Indeed, any secreted thrombin activity would likely render the anchors nonfunctional. Furthermore, while some cells do not bind to fibrinogen, such as S2 cells (Fig. 5), fibrinogen has been widely used to functionalize surfaces to allow cell adhesion. If cells just bind to fibrinogen as they would bind to extracellular matrix proteins (like fibronectin), then using fibrinogen as an anchor to micropattern ligands asymmetrically, like we do in Fig. 6 and Fig. 7, would actually be an advantage, as the density of adhesion sites could be kept identical between the fibrinogen and fibrinogen-ligand part of the micropatterns. However, if one wants to investigate cell mechanics by functionalizing fibrinogen with force sensors (LaCroix et al., 2018; Blakely et al., 2014) or light-controlled force inducers (Zheng et al., 2020 *Preprint*), then the direct binding of the cell to the fibrinogen itself rather than the force-sensing/inducing moiety would be an issue, as it will interfere with the experiments. Another potential concern for these experiments is that if cells are able to bind to fibrinogen (or fibrinogen anchors), then strong enough cells might be able to rip off the micropatterns over time, as was shown previously with fibronectin (Fink et al., 2007). Indeed, as good as fibrinogen micropatterns are, they are still achieved by protein adsorption. While we did not observe this in any of the experiments described in this paper, as all fibrinogen micropatterns presented in this study were stable when interacting with 3T3, HeLa, and U2OS cells (Fig. 6 and Fig. 7), this might be a concern for researchers working with other cells and/or much longer experiments. A solution to this potential problem could be to use mild aldehyde or N-hydroxysuccinimide (NHS) fixatives to cross-link the fibrinogen molecules of the micropattern before adding cells. Indeed, fibrinogen functionalized in the conditions described here still offers reactive amines, a property we exploited to generate bi-functional fibrinogens like fibrinogen-biotin-ATTO490LS. Conversely, one could also tune the interaction between the anchor and the cell, using weaker nanobodies for instance, so the anchor would unbind when the cell pulls too much on it. These potential caveats notwithstanding, now that we have identified the key properties that make a good micropatterning anchor, it would be an interesting avenue of future research to find an alternative to fibrinogen that has no biological activity.

The combination of all the advantages offered by our technology enabled us to achieve subcellular micropatterning of proteins, whereby the position of proteins can be imprinted from the outside via a micropattern (Fig. 6 and Fig. 7). This was not possible with direct micropatterning of ligands (Fig. S5, B and C), demonstrating the importance of the fibrinogen toolbox described here. While beyond the scope of this paper, we envision that this technology will allow researchers to untangle the interplay between mechanical forces (provided by the distribution of adhesion molecules) and signaling (provided by the distribution of signaling receptors). Conversely, by trapping endogenous receptor-ligand complexes into the total internal reflection fluorescence (TIRF) field while blocking their endocytosis, this assay could help the characterization of the sequence of recruitment of proteins at endocytic sites, which could be then structurally characterized thanks to the recent combination of CryoEM and micropatterning (Toro-Nahuelpan et al., 2020; Engel et al., 2019). Furthermore, the reconstitution of signaling platforms of defined receptor-ligand content at a known density is poised to help decipher the combinatorial interactions between signaling pathways, such as the cross-talk between EGFR and Her2, which has been proposed to underlie resistance to anticancer drugs targeting these receptors (Yamaguchi et al., 2014). Last, the orthogonal transmembrane segment we established (Fig. 6) could be used to relocalize other, intracellular proteins to the cell cortex in a controllable fashion via a micropattern, an assay reminiscent of what we previously achieved in vivo using polarity markers as an anchoring platform (Derivery et al., 2015). This could, for instance, be used to generate symmetry breaking of the actin cortex via asymmetric targeting of cytoskeleton regulators (by fusing cytoskeleton regulators to the transmembrane construct, for instance).

In conclusion, we hope that the advantages of the fibrinogen anchors described in this paper will be a stepping-stone toward having micropatterning experiments limited only by the imagination of the researcher rather than by intrinsic limitations of this powerful technology.

## Materials and methods
### Synthesis of the photosensitizer BBTB
All chemical reagents were purchased with the highest purity available. Briefly, in a 100-ml round-bottom flask, 2.1 ml trimethylamine solution 4.2 M in ethanol (1.2 equivalents, 8.8 mmol; Sigma-Aldrich) was added to 2.0 g of 4-(bromomethyl) benzophenone (1 equivalent, 7.3 mmol; Sigma-Aldrich) resuspended in 40 ml acetonitrile. The reaction was placed under reflux for 2 h in a nitrogen atmosphere. Once the reaction was completed, the solvent was evaporated, and the white solid obtained was dried under vacuum (2.35 g, 97% yield).

Characterization of the product: high-performance liquid chromatography (HPLC): 95% pure; $^1$H nuclear magnetic resonance (400 MHz, deuterated methanol) δ: 7.93 (m, 2H), 7.84 (m, 2H), 7.77 (m, 2H), 7.70 (m, 1H), 7.58 (m, 2H), 4.71 (s, 2H), 3.22 (s, 9H). $^{13}$C nuclear magnetic resonance (400 MHz, deuterated methanol) δ: 139.5, 136.8, 132.8, 132.8, 131.7, 130.0, 129.7, 128.3, 68.3, 52.0; mass spectrometry (electrospray ionization) $[m-Br]^+$/z calculated for $C_{17}H_{20}NO$: 254.1, found: 254.1.

## Plasmids

The GBP3xCys consists of a nanobody against GFP (Rothbauer et al., 2008), referred to as GBP, with three cysteines added far from the active site to minimize loss of activity upon functionalization through these cysteines (Pleiner et al., 2015). GBP3xCys was amplified by PCR from the P434_H14-Sp-brNEDD8-Cys-anti-GFP nanobody (a kind gift from Tino Pleiner, Max Planck Institute for Biophysical Chemistry, Göttingen, Germany) and cloned into a modified pRSET vector (a kind gift from Mark Allen, Medical Research Council Laboratory of Molecular Biology, Cambridge, UK) tagging N-terminally the ORF with a $(His)_6$ tag, followed by the first 93 amino acids of the dihydrolipoyl acetyltransferase component of pyruvate dehydrogenase complex from *Bacillus stearothermophilus* to enhance solubility (as previously described by Watson et al., 2020), and a tobacco etch virus (TEV) cleavage site (referred to as His-SS-TEV). Primers used to amplify GBP3xCys were Fse_gbp_3cys: 5′-ATGCGGCCG GCCCTGCGGATCCCAGGTACAGCT-3′ and Asc_gbp_3cys: 5′-ATGCGGCGCGCCCTTAACACCTAGTACCGGAGCTGC-3′. EGFP was PCR-amplified from pEGFP-C2 (Clontech) and similarly cloned into the pRSET His-SS-TEV vector using the following primers: Fse_GFP_pRSET: 5′-ATGCGGCCGGCCCATGAGCAA GGGCGAGGAGCTGTT-3′ and Asc_GFP_pRSET: 5′-ATGCGG CGCGCCCTTACTTGTACAGCTCGTCCATGC-3′.

pWC2 (Subramanian and Gelles, 2007) encoding the motor domain of *Drosophila* kinesin 1 fused to the biotin carboxyl carrier protein noted Kin1-biotin was a gift from Jeff Gelles (Department of Biochemistry, Brandeis University, Waltham, MA; Addgene plasmid no. 15960; http://n2t.net/addgene:15960; Research Resource Identifier: Addgene_15960).

For the purification of streptavidin-GFP-GFP, two GFPs were cloned into the pRSET His-SS-TEV-streptavidin vector, which permits solubilization of natively purified streptavidin (Watson et al., 2020). The GFP sequences were codon optimized to prevent repetitive sequences. The resulting template was synthesized by GeneArt (Thermo Fisher Scientific) and PCR-amplified using the following primers: Fse_2XGFP_f: 5′-ATGCGGCCGGCC CATGGTGAGCAAGGGCGAGGA-3′ and Asc_2XGFP_r: 5′-ATG CGGCGCGCCCCTAATGATGATGATGATGATGTTTATACAATT CATCCATAC-3′.

The transmembrane nanobody construct (Fig. 6) comprises an N-terminal signal peptide from the *Drosophila* echinoid protein, followed by $(His)_6$–Protein C tandem affinity tags, the GBP nanobody against GFP (Rothbauer et al., 2008), a TEV protease cleavage site, the transmembrane domain from the *Drosophila* echinoid protein, the vesicular stomatitis virus G export sequence, and the mScarlet protein (Bindels et al., 2017). The protein expressed by this construct thus consists of an extracellular anti-GFP nanobody linked to an intracellular mScarlet by a transmembrane domain (named GBP-TM-mScarlet in the main text for simplicity). This custom construct was synthesized by IDT and cloned into a modified pCDNA5/FRT/V5-His vector (Thermo Fisher Scientific), where the cytomegalovirus promoter was replaced by the EF1α/human T-lymphotropic virus chimera promoter from the pFUSE fragment crystallizable region vector (pFUSE Fc; Invivogen). This modification did not affect the ability of the plasmid to undergo homologous recombination into the FRT site (FlipIn system; Thermo Fisher Scientific). The C-terminal V5-His tandem tag present in the plasmid is not translated because of a stop codon introduced at the end of the mScarlet sequence in the synthesized GBP-TM-mScarlet sequence.

## Proteins

Unless stated otherwise, all protein purification steps were performed at 4°C. Protein concentration was determined either by absorbance at 280 nm or by densitometry on a Coomassie-stained SDS page gel against a BSA ladder.

GBP3xCys, streptavidin-GFP-GFP, and GFP were expressed in *Escherichia coli* BL21 Rosetta 2 (Stratagene) using the plasmids described above by induction at OD 600 = 0.8 with 1 mM IPTG in 2× yeast triptone (streptavidin-GFP-GFP) or terrific broth (GFP, GBP3xCys) mediua at 20°C. Bacteria were lysed by sonication in lysis buffer (20 mM Hepes, 150 mM KCl, 1% Triton X-100, 5% glycerol, 20 mM imidazole, 5 mM $MgCl_2$, and 0.5 mM DTT, pH 7.6) enriched with protease inhibitors (Roche Mini) and 1 mg/ml lysozyme (Sigma-Aldrich) and 10 μg/ml DNase I (Roche). After clarification (16,000 rpm, Beckman JA 25.5, 30 min, 4°C), lysate was incubated with nickel-chelating nitrilotriacetic acid (Ni-NTA) resin (Qiagen) for 2 h at 4°C and washed extensively in 20 mM Hepes, 150 mM KCl, 5% glycerol, 15 mM imidazole, and 0.5 mM DTT, pH 7.6. Protein was eluted in 20 mM Hepes, 150 mM KCl, 5% glycerol, 250 mM imidazole, and 0.5 mM DTT, pH 7.6. Bradford-positive elution fractions were pooled and TEV-cleaved overnight by adding 1:50 (vol:vol) of 2 mg/ml $(His)_6$-TEV protease and 1 mM/0.5 mM final DTT/EDTA. The solution was then dialyzed twice against 20 mM Hepes, 150 mM KCl, 15 mM imidazole, and 0.5 mM DTT, pH 7.7, and TEV protease and His-SS-TEV were removed using Ni-NTA resin. Tag-free GBP3xCys was then concentrated down to 10 mg/ml, flash frozen in liquid $N_2$, and kept at −80°C. GFP and streptavidin-GFP-GFP were similarly purified, except that DTT was omitted in the lysis and storage buffers, and that the final concentration was 4.22 mg/ml (GFP) and 0.75 mg/ml (streptavidin-GFP-GFP).

BSA-LC-LC-biotin-Alexa647 (BSA-biotin-Alexa647) was generated by reacting 30 μM BSA (BP1605; Thermo Fisher Scientific; dissolved in 0.1 M sodium bicarbonate, pH 8.3) simultaneously with a fivefold molar excess of Alexa647 NHS ester (A20006; Thermo Fisher Scientific) and a fivefold molar excess of EZ-Link NHS LC-LC-biotin (21343; Thermo Fisher Scientific). Both NHS-reactive chemicals were resuspended immediately before the reaction in a new vial of anhydrous DMSO (D12345; Invitrogen). The reaction was rocked for 1 h at RT before removal of unreacted Alexa647 and biotin on a Zeba Spin column (89891; Thermo Fisher Scientific) equilibrated in 0.1 M sodium bicarbonate, pH 8.3. The degree of Alexa647 labeling was 1.48 mol of dye per mol of BSA, and the resulting BSA-biotin-Alexa647 visually bound to agarose beads conjugated to streptavidin (20359; Pierce).

To generate streptavidin-Alexa647, streptavidin (1 mg/ml in PBS plus 60 mM sodium bicarbonate, pH 8.0) was reacted with a sixfold molar excess of Alexa647 NHS ester (A20006; Thermo Fisher Scientific). Excess dye was removed by Zeba Spin column

equilibrated in 20 mM Hepes, 150 mM KCl, and 5% glycerol, pH 7.6.

Kin1-biotin was expressed using the plasmid pWC2 by induction at OD 600 = 0.8 with 1 mM IPTG in 2× yeast triptone medium at 20°C. Bacteria were lysed by sonication in lysis buffer (20 mM Hepes, 150 mM KCl, 1% Triton X-100, 5% glycerol, 0.1 mM ATP, 10 mM imidazole, 10 mM MgCl$_2$, and 1 mM DTT, pH 7.6) enriched with protease inhibitors (Roche Mini) and 0.7 mg/ml lysozyme (Sigma-Aldrich) and 10 µg/ml DNase I (Roche). After clarification (14,000 rpm, 30 min, 4°C; Beckman JA 25.5), lysate was incubated with Ni-NTA resin (Qiagen) for 2 h at 4°C and washed extensively in wash buffer (20 mM Hepes, 150 mM KCl, 5% glycerol, 0.1 mM ATP, 10 mM imidazole, 2 mM MgCl$_2$, and 1 mM DTT, pH 7.6), followed by wash buffer enriched with 2 mM ATP and a final wash in wash buffer. Protein was eluted in 20 mM Hepes, 150 mM KCl, 5% glycerol, 0.1 mM ATP, 300 mM imidazole, 2 mM MgCl$_2$, and 1 mM DTT, pH 7.6. Bradford-positive elution fractions were pooled and diluted 1:11 (vol eluate:vol QA) in buffer QA (20 mM Hepes, 20 mM KCl, 1 mM MgCl$_2$, 0.05 mM ATP, and 1 mM DTT, pH 7.6) before loading onto an MonoQ 5/50 GL anion exchange column (0.5 ml/min). Protein was then eluted in a 0.05–1 M gradient of KCl. Positive fractions were pooled, dialyzed against storage buffer (20 mM Hepes, pH 7.7, 150 mM KCl, 1 mM MgCl$_2$, 0.05 mM ATP, 1 mM DTT, and 20% (wt/vol) sucrose, pH 7.6), flash frozen in liquid N$_2$, and kept at −80°C (final concentration, 2.2 mg/ml).

NeutrAvidin, NeutrAvidin DyLight-550, and Fibrinogen-Alexa546 were purchased from Thermo Fisher Scientific. Unlabeled fibrinogen was purchased from MP Biomedicals (08820224).

Tubulin, HiLyte-647 tubulin, and biotinylated tubulin were all purchased from Cytoskeleton. GMPCPP-stabilized MTs were polymerized by resuspending 1-µl aliquots of HiLyte-647–labeled tubulin mixes (60% unlabeled tubulin, 20% HiLyte-647 tubulin, 20% biotinylated tubulin; T240, TL670M, and T333, respectively; Cytoskeleton, Inc.) in 50 µl of warm solution (80 mM Pipes, 2 mM MgCl$_2$, 0.5 mM EGTA, and 0.6 mM GMPCPP, pH 6.9), before incubation for 30 min at 37°C. MTs were pelleted for 8 min at 15,871 $g$ and resuspended in 50 µl of 80 mM Pipes, 2 mM MgCl$_2$, 0.5 mM EGTA, and 0.6 mM GMPCPP, pH 6.9.

Delta-like ligand 4 (DLL4), comprising a fragment of the human Delta ectodomain (1–405) with a C-terminal GS-SpyTag-His$_6$ sequence, was purified from culture medium of transiently transfected Expi293F cells (Thermo Fisher Scientific) by metal affinity chromatography. Protein was further purified by size exclusion chromatography on a Superdex 200 column in 50 mM Tris, pH 8.0, 150 mM NaCl, and 5% glycerol. DLL4 (69 µM) was subsequently buffer exchanged into 0.1 M sodium bicarbonate, pH 8.3, using a Zeba Spin column and labeled with a threefold molar excess of EZ-Link NHS LC-LC-biotin (Thermo Fisher Scientific) for 1 h at RT. Unreacted biotin was then removed by Zeba Spin column equilibrated in 0.1 M sodium bicarbonate, pH 8.3.

## Fibrinogen fusions
### Three-step synthesis of fibrinogen-NeutrAvidin, fibrinogen GFP, and fibrinogen–Con A
Fibrinogen powder was resuspended at 4 mg/ml (~11.8 µM) in fibrinogen buffer (0.1 M sodium bicarbonate, 0.5 mM EDTA, pH 8.3) before addition of a 25-fold molar excess of Traut's Reagent

(2-iminothiolane; Pierce). The solution was gently rocked for 45 min at RT. 1 mM DTT final concentration was then added, and the solution was further rocked for 1 h at RT. Excess DTT/Traut's reagent was then removed using a Zeba Spin column equilibrated in fibrinogen buffer. Concomitantly, the buffer of purified NeutrAvidin (80 µM) was exchanged for fresh fibrinogen buffer using a Zeba Spin column, and then a sixfold molar excess of maleimide-PEG8-succinimidyl ester (746207; Sigma-Aldrich) was added. After rocking for 1 h at RT, the excess maleimide-PEG8-succinimidyl ester was removed on a Zeba Spin column equilibrated in fibrinogen buffer. NeutrAvidin-maleimide was then added to fibrinogen-thiol in a fourfold NeutrAvidin-maleimide/fibrinogen molar ratio and rocked overnight at 4°C. 50 mM final free cysteine was then added and incubated for 30 min at RT to quench the remaining unreacted maleimide functions. The solution was then brought to 25% final ammonium sulfate by adding one third of the volume of saturated ammonium sulfate to specifically precipitate the fibrinogen-NeutrAvidin. After 30 min rocking at RT, the solution was centrifuged at 16,873 $g$ for 10 min. The pellet was resuspended in the initial volume of fibrinogen buffer by gentle rocking for 1 h at 4°C, followed by a second precipitation in 25% final ammonium sulfate as above. After centrifugation at 16,873 $g$ for 10 min, the pellet was resuspended in one quarter of the initial volume of fibrinogen buffer, ultracentrifuged to remove aggregates (100,000 $g$, 5 min at 4°C), then flash frozen in liquid N$_2$ and kept at −80°C.

Fibrinogen-GFP was similarly obtained by replacing NeutrAvidin by purified GFP, and, in this case, we used a 15-fold molar excess of GFP-maleimide over fibrinogen-thiol. The absorbance at 488 nm of GFP allowed us to evaluate the degree of labeling to 0.51 mol of GFP per mol of fibrinogen in these conditions.

Fibrinogen–Con A was similarly obtained by replacing NeutrAvidin with purified Con A (Sigma-Aldrich).

### Two-step synthesis of fibrinogen-GBP
Fibrinogen powder was resuspended at 4 mg/ml (~11.8 µM) in fibrinogen buffer (0.1 M sodium bicarbonate and 0.5 mM EDTA, pH 8.3), and then a 25-fold molar excess of maleimide-PEG8-succinimidyl ester was added. After rocking for 1 h at RT, the excess maleimide-PEG8-succinimidyl ester was removed on a Zeba Spin column equilibrated in fibrinogen buffer. Concomitantly, the buffer of purified GBP3xCys was exchanged for fresh fibrinogen buffer using a Zeba Spin column, which also removed the DTT used in the purification of the GBP3xCys to keep cysteines reduced. GBP3xCys (10 mg/ml) was then added to fibrinogen-maleimide in a fivefold GBP::fibrinogen-maleimide molar ratio and rocked overnight at 4°C. 50 mM (final) free cysteine was then added and incubated for 30 min at RT to quench the remaining unreacted maleimide functions. Then the solution was brought to 25% final ammonium sulfate as above to specifically precipitate the fibrinogen-GBP. After 30 min rocking at RT, the solution was centrifuged at 16,873 $g$ for 10 min, and the pellet was resuspended in the initial volume of fibrinogen, followed by a second precipitation in 25% final ammonium sulfate. After centrifugation at 16,873 $g$ for 10 min, the pellet was resuspended in one quarter of the initial volume of fibrinogen

buffer, ultracentrifuged to remove aggregates (100,000 $g$, 5 min at 4°C), then flash frozen in liquid $N_2$ and kept at –80°C.

### One-step synthesis of fibrinogen-biotin, fibrinogen-Alexa647, fibrinogen-ATTO488, fibrinogen-biotin-ATTO490LS, and fibrinogen-biotin-Alexa647

Fibrinogen-biotin was obtained by mixing fibrinogen (4 mg/ml in fibrinogen buffer: 0.1 M sodium bicarbonate and 0.5 mM EDTA, pH 8.3) with a 20-fold molar excess of EZ-Link NHS LC-LC-biotin (Pierce) for 1 h at RT. Unreacted biotin was then removed by dialysis against fibrinogen buffer (2 × 2 liters).

Fibrinogen-Alexa647 and fibrinogen-ATTO488 were obtained by reacting fibrinogen (4 mg/ml in fibrinogen buffer) with a 10-fold molar excess of NHS-Alexa647 (A20006; Thermo Fisher Scientific) or NHS-ATTO 488 (ATTO-TECH AD 488–31), respectively, for 1 h at RT. Unreacted dye was then removed by Zeba Spin column equilibrated in fibrinogen buffer.

Fibrinogen-biotin-ATTO490LS and fibrinogen-biotin-Alexa647 were generated by mixing fibrinogen (4 mg/ml in fibrinogen buffer) with a threefold molar excess of ATTO490LS NHS ester (ATTO-TEC) or Alexa647 NHS ester (A20006; Thermo Fisher Scientific) for 15 min at RT. The fibrinogen-ATTO490LS and fibrinogen-Alexa647 were subsequently reacted with a 50-fold molar excess of EZ-Link NHS LC-LC-biotin for 1 h at RT. Unreacted biotin and fluorescent dye were then removed by Zeba Spin column equilibrated in fibrinogen buffer.

### Cells

*Drosophila* S2 cells (mycoplasm-free judged by DAPI staining; University of California, San Francisco) were grown in Schneider Medium (Gibco) supplemented with 10% heat-inactivated FBS. Flp-In NIH/3T3 cells (Invitrogen) were cultured in DMEM (Gibco) supplemented with 10% Donor Bovine Serum (Gibco) and Pen/Strep 100 units/ml at 37°C with 5% $CO_2$. HeLa cells (CCL-2; American Type Culture Collection) were cultured in DMEM supplemented with 10% FBS (Gibco) and Pen/Strep 100 units/ml at 37°C with 5% $CO_2$. U2OS cells (HTB-96; ATCC) stably expressing inducible FLAG-Notch1-EGFP chimeric receptors (Malecki et al., 2006) were maintained in DMEM supplemented with 10% FBS, Pen/Strep 100 units/ml, 50 µg/ml hygromycin B (Thermo Fisher Scientific), and 15 µg/ml blasticidin (Invitrogen) at 37°C with 5% $CO_2$. Prior to use in experiments, U2OS cells were induced with 2 µg/ml doxycycline for 24 h. Flp-In NIH/3T3 were transfected with a modified pCDNA5 vector containing GBP-TM-mScarlet with lipofectamine 2000 (Invitrogen). Stable transfectants by homologous recombination at the FRT site were obtained according to the manufacturer's instructions by selection with 100 µg/ml Hygromycin B Gold (Invivogen). All live imaging was performed in Leibovitz's L-15 medium (11415064; GIBCO BRL) supplemented with 10% Donor Bovine Serum and Hepes (20 mM; 1563080; GIBCO BRL).

### Immunofluorescence

Cells were fixed 4% paraformaldehyde in PBS for 20 min, permeabilized with 0.1% Triton X-100 in PBS for 5 min, then washed in PBS, then in PBS supplemented with 1% BSA and 1% rabbit IgG blocking reagent (Thermo Fisher Scientific) for 5 min,

then in PBS. Anti-clathrin (ab21679; Abcam) and anti-EGF receptor (D38B1; Cell Signaling) were labeled with Alexa488- and Alexa647-Zenon rabbit IgG labeling kits, respectively (Thermo Fisher Scientific), according to the manufacturer's instructions, and cells were then incubated with both antibodies (both diluted 1:100 in PBS–1% BSA) for 20 min. After washing thrice in PBS, imaging was performed in PBS instead of mounting medium to avoid squashing the cells (and potentially biasing the clathrin or EGFR micropattern colocalization).

### Microscopy

Imaging was performed on a custom TIRF/spinning-disk confocal microscope composed of a Nikon Ti stand equipped with perfect focus and a 100× NA 1.45 Plan Apochromat lambda objective (or alternatively, a 60× NA 1.49 Apochromat TIRF or a 10× NA 0.3 Plan Fluor). The confocal imaging arm is composed of a Yokogawa CSU-X1 spinning-disk head and a Photometrics 95B back-illuminated sCMOS camera operating in global shutter mode and synchronized with the spinning-disk rotation. Conversely, the TIRF imaging arm is composed of an azimuthal TIRF illuminator (iLas2; GATACA Systems) modified to have an extended field of view (Cairn) to match the full field of view of the camera. Images are recorded with a Photometrics Prime 95B back-illuminated sCMOS camera run in pseudo-global shutter mode and synchronized with the azimuthal illumination. Excitation is performed using 488- (150 mW OBIS LX), 561- (100 mW OBIS LS), and 637-nm (140mW OBIS LX) lasers fibered within a Cairn laser launch. To minimize bleed-through, single-band emission filters are used (Chroma 525/50 for GFP/ATTO488, 595/50 for Alexa546, and ET655lp for Alexa647/ATTO647N/ATTO490LS/SiR-tubulin), and acquisition of each channel is performed sequentially using a fast filter wheel (Cairn Optospin) in each imaging arm. Filter wheels also contain a quad-band filter (Chroma ZET405/488/561/640m) to allow imaging of the reflection of the DMD illumination arm at the glass/water interface (see below). To enable fast acquisition, the entire setup is synchronized at the hardware level using an field-programmable gate array (FPGA) stand-alone card (sbRIO 9637; National Instrument) running custom code. TIRF angle was set independently for all channels so that the depth of the TIRF field was identical for all channels. Sample temperature was maintained at 25°C using a heating enclosure (https://MicroscopeHeaters.com). Acquisition was controlled by Metamorph software.

### Optical design of a low-cost UV light-emitting diode (LED) DMD illuminator

Our optical design combines DMD-UV illumination with TIRF illumination onto the Nikon Ti setup described above (Fig. S1 A). Briefly, a 385-nm high-power UV LED light source (M385LP1; Thorlabs) is collimated using an antireflective (AR)-coated aspheric lens (ACL2520U-A; Thorlabs). The collimated UV beam is then directed toward a DMD (DLPLCR6500EVM; Texas Instruments) at a 24° angle of incidence (corresponding to twice the tilting angle of the DMD mirrors). The image of the DMD chip is then relayed onto the conjugate of the sample plane at the backport of the microscope through a 4f imaging system (f1 = f2 = 125 mm UV Fused Silica Bi-Convex Lenses, AR-Coated,

LB4913-UV; Thorlabs). This intermediate image is then relayed onto the sample plane by a tube lens (125 mm, UV rated; Edmond Optics) and the objective (100× Plan Apochromat lambda NA 1.45, Plan Apochromat 60× NA 1.4, or 20× Plan Apochromat Lambda VC NA 0.75). To combine DMD-UV illumination with TIRF illumination, an ultraflat dichroic (T470lpxr; Chroma) is placed after f2 within a custom backport assembly (Cairn). When in DMD-UV illumination mode, the microscope filter cube turret contains a 473-nm dichroic (Di03-R473; Semrock), while it contains an ultraflat quad-band dichroic/clean-up filter (TRF89901-EM; Chroma) when in TIRF illumination mode. The 473-nm dichroic is compatible with simultaneous spinning-disk/DMD illumination. We note that care must be taken with the adjustment of the collimating lens of the LEDs to find the best compromise between illumination intensity and flatness of the illumination profile. If necessary, a flatfield correction can be applied on the micropatterns to be displayed to account for any field inhomogeneities, and if extremely sharp micropatterns are required, an iris can be put in the Fourier plane between f1 and f2, but we found that this was not required for most applications.

To offer a second illumination wavelength for 450-nm optogenetic stimulation using the same DMD chip, our design also contains a second collimated LED (450 nm; M450LP1; Thorlabs, in our case) at the symmetric –24° angle. Therefore, any micropattern can be displayed using the 450-nm light source by simply inverting the micropattern before displaying it on the DMD. This could be replaced by any other light source to bring epifluorescence imaging in order to facilitate multiprotein micropattern alignment on a setup not equipped with TIRF.

LED intensity is controlled using a custom LED driver providing the maximum 1.7 A tolerated by the LED (2 A for 450 nm LED). Control over LED intensity and on/off state is operated using a digital/analogue card (USB-6001; National Instrument; or Arduino UNO equipped with a custom shield providing a Texas Instrument TLV5618 digital/analogue chip). Communication to the DMD from the imaging software is performed using the DMD Connect library developed by Hueck (2016), available at https://github.com/deichrenner/DMDConnect. Control of all parts was integrated into Metamorph and/or Micromanager using custom scripts to calibrate the DMD with respect to the camera, display user-defined UV micropatterns, and facilitate micropattern alignment for multiprotein micropatterning.

To keep the cost low and to enable researchers with limited access to mechanical workshops to make their own module, our design relies on a commercially available cage system to set the +24° angle and a custom mount for the DMD board, which can be machined or 3D printed. This greatly facilitates alignment of the setup, as this essentially locks all pieces into the correct angle during assembly, which is critical for alignment of DMD setups (Strale et al., 2016). All codes and computer-aided design files for this setup are available freely upon request for noncommercial purposes.

Importantly, efficient and crisp micropatterning requires that the microscope is focused on the PLL-PEG (or PEG-silane)/glass interface. While hardware autofocus systems help in finding this interface, they do not always work perfectly as there is usually a correction offset to add, which may vary from sample to sample. To ensure that we always focus the instrument at the right place, we use the fact that because micropatterning is performed in aqueous buffer, there is a glass/liquid interface at the PLL-PEG layer, which reflects the UV micropatterning light. We thus chose our filter/dichroic sets to allow some of this reflected light to be imaged onto the camera, which allows us to easily find the optimal sample plane in the absence of anything fluorescent in the chamber. Once this plane has been found, we activate the hardware autofocus system to ensure that this plane is kept during micropatterning process.

**Protein micropatterning**
For micropatterning on PLL-PEG–passivated coverslips in an open configuration (all micropatterns except Fig. S3), clean room–grade coverslips (custom 25 × 75–mm size; Nexterion) were surface activated under pure oxygen in a plasma cleaner (PlasmaPrep2; GaLa Instruments) and then laid on top of a 200-µl drop of filtered PLL(20)-g[3.5]-PEG(2) (PLL-PEG; 100 µg/ml in 10 mM Hepes, pH 7.6; SuSoS) for 1 h in a humid chamber. Coverslips were then washed extensively in filtered MilliQ water and dried under a flow of dry nitrogen gas. Coverslips were then mounted in a sticky slide Ibidi 8-well chamber (80828; Ibidi) and pressed under a ~4-kg weight overnight to stick the chamber to the glass. Then, per well of the 8-well chamber, 200 µl of BBTB was added (50 mM in 0.1 M sodium bicarbonate, pH 8.3) and exposed to micropatterned UV light using the DMD-UV arm of the micropatterning microscope (30–90-s exposure) after focusing on the glass/buffer interface. BBTB was subsequently removed by repeated dilution (12 washes with 600 µl of 0.1 M sodium bicarbonate, pH 8.3) before addition of 200 µl of fibrinogen anchor (100 µg/ml in 0.1 M sodium bicarbonate, pH 8.3, for a final concentration of 50 µg/ml). The fibrinogen anchor was allowed to adsorb to the surface for 5 min before washing by repeated dilution (12 washes). The surface was then quenched for 5 min with 0.5 mg/ml PLL-PEG (in 10 mM Hepes, pH 7.6). PLL-PEG was then removed by similar repeated dilution, so that the surface never dried. The protein of interest was then added in its buffer of choice (here 0.1 M sodium bicarbonate, pH 8.3, for GFP, biotin-BSA, and NeutrAvidin) at 5 µg/ml and allowed to bind to the fibrinogen anchor for 5 min before extensive washes and imaging. Alternatively, when proteins were directly micropatterned, the fibrinogen anchor in the above protocol was replaced by the protein of interest at a concentration of 50 µg/ml in 0.1 M sodium bicarbonate, pH 8.3 (except for fluorescent EGF, where biotinylated EGF complexed to streptavidin-Alexa555 [E35350; Invitrogen] was patterned at 1 µg/ml in 0.1 M sodium bicarbonate, pH 8.3). After a 5-min incubation, the surface was quenched and washed as above.

For sequential multiplexed micropatterning experiments on PLL-PEG–passivated coverslips (Fig. 1, C–E; Fig. 3; and Fig. S1, D–F), the protocol above was performed except that immediately after the PEG quenching step, BBTB was added to the chamber (50 mM in 0.1 M sodium bicarbonate, pH 8.3), and the whole process was repeated for a second fibrinogen anchor (and a third time for the triple patterning in Fig. 1, C–E). For Fig. 3, where fibrinogen

anchors were used to subsequently bind to two proteins of interest, said proteins of interest were added together after micropatterning of both fibrinogen anchors (5 µg/ml each in 0.1 M sodium bicarbonate, pH 8.3, for 5 min) before extensive washes and imaging. All fibrinogen anchors that are not fluorescent were doped with 5 µg/ml fibrinogen-Alexa546 to image their respective micropattern and thereby align the next micropattern. For controls where one fibrinogen anchor was omitted, it was replaced with fibrinogen-Alexa546.

For dual micropatterning experiments where receptors were clustered to micropatterns (Fig. 6, Fig. 7, and Fig. S5), the above protocol was performed, with the small central region micropatterned first (50 µg/ml GFP, fibrinogen-GFP, fibrinogen-biotin-ATTO490LS, or fibrinogen-biotin-Alexa647), before PLL-PEG quenching and micropatterning of the second, outer region (50 µg/ml fibronectin plus 10 µg/ml fibrinogen-Alexa546 or fibrinogen-Alexa647). For relocalization of the GBP-TM-mScarlet receptor, micropatterns (with a central fibrinogen-biotin-ATTO490LS center) were incubated (or not) with streptavidin-GFP-GFP (10 µg/ml) and fixed with 0.5 mM dithiobis(succinimidyl propionate) for 20 min. After extensive washing, NIH/3T3 cells were added (20,000 cells/well) in serum-free DMEM for 30 min before washing into medium containing serum and imaging 30 min later. Alternatively, for live imaging of GBP-TM-mScarlet recruitment (Fig. 6), 20,000 cells were added per well, in L15 plus 20 mM Hepes, and imaged as they landed and spread on micropatterns. For relocalization of EGFR and GFP-Notch1, double micropatterns were incubated with NeutrAvidin (25 µg/ml) for 5 min before extensive washing and incubation for 5 min with biotinylated-EGF (1 µg/ml) or biotinylated-DLL4 (0.5 µM), respectively. Chambers were then extensively washed. HeLa cells were serum starved for 24 h before addition, in serum-free medium, to EGF micropatterns (20,000 cells/well). Cells were left to spread for 40 min before fixation. U2OS GFP-Notch1 cells, induced for 24 h to express GFP-Notch1, were added to DLL4 micropatterns in L15 plus 20 mM Hepes (20,000 cells/well) and imaged as they landed and spread on micropatterns.

For micropatterning of Kin1-biotin using the fibrinogen-biotin::NeutrAvidin "sandwich" (Fig. 4), PLL-PEG–passivated coverslips assembled into 8-well chambers were exposed for 90 s before the addition of fibrinogen-biotin-ATTO490LS (50 µg/ml in 0.1 M sodium bicarbonate, pH 8.3) as above. Micropatterns were then quenched with 0.5 mg/ml PLL-PEG for 5 min before addition of NeutrAvidin (10 µg/ml in 0.1 M sodium bicarbonate, pH 8.3, for 5 min) before washing and incubation with 5 µg/ml purified Kin1-biotin in 80 mM Pipes, 1 mM MgCl$_2$, and 5 mM ATP, pH 8.9, for 5 min. Alternatively, for direct micropatterning of Kin1-biotin, micropatterns were exposed to micropatterned UV light in the presence of BBTB for 90 s as above before addition of 5 µg/ml Kin1-biotin in 80 mM Pipes, 1 mM MgCl$_2$, and 5 mM ATP, pH 8.9. Micropatterns were then quenched with 0.5 mg/ml PLL-PEG for 5 min. 4 µl of GMPCPP MT seeds (see above) were then added to each well, in 200 µl of ATP-regenerating buffer (80 mM Pipes, 0.1 mg/ml κ-casein, 40 µM DTT, 64 µM glucose, 160 µg/ml glucose oxidase, 20 µg/ml catalase, 0.1% methylcellulose, and 1 mM ATP, pH 6.9), and

imaged by TIRF microscopy. For direct micropatterning of Kin1-biotin, visualization of micropatterns was performed after imaging by addition of 1 µM streptavidin-Alexa647.

For micropatterning of Con A to bind to S2 cells (Fig. 5), surface-activated coverslips were passivated with 0.1 mg/ml PLL-PEG before being micropatterned using a quartz-chromium photomask and a deep UV light source for 5 min as described previously (Azioune et al., 2009). Exposed coverslips were then incubated with 200 µl fibrinogen alone (50 µg/ml fibrinogen and 5 µg/ml fibrinogen-Alexa546), Con A alone (50 µg/ml Con A and 5 µg/ml rhodamine–Con A), or fibrinogen–Con A (50 µg/ml fibrinogen–Con A and 5 µg/ml fibrinogen-Alexa546), all in 0.1 M sodium bicarbonate, pH 8.3, for 1 h. Coverslips were washed in 0.1 M sodium bicarbonate, pH 8.3, and assembled into an 8-well chamber. 100,000 S2 cells in low (1%) serum media were then added to each well for 1 h before incubation with SiR-tubulin (1 µM; Spirochrome; Lukinavičius et al., 2014) for 30 min to label cells. Chambers were washed with Schneider medium plus 10% heat-inactivated FBS and imaged by confocal spinning-disc microscopy.

For micropatterning on PEG-silane–passivated coverslips in a flow cell configuration (Fig. S3), clean room–grade coverslips (75 × 25 mm; Nexterion) were surface-activated under pure oxygen in a plasma cleaner and then incubated with PEG-silane (30 kD; PSB-2014; Creative PEGWorks) at 1 mg/ml in ethanol 96%/0.1% HCl overnight at RT with gentle agitation. Standard 22 × 22–mm coverslips were similarly passivated with PEG-silane, omitting the plasma-cleaning step. Slides and coverslips were then successively washed in 96% ethanol and ultrapure water before drying under nitrogen gas. The coverslip and slide were subsequently assembled into an array of six flow cells (~15 µl each) using double-sided tape (AR-90880; Adhesive Research; cut with a Graphtec CE6000 cutting plotter). The flow cell chamber was then filled with BBTB (50 mM in 0.1 M sodium bicarbonate, pH 8.3) and exposed to UV light on the DMD-UV arm of the micropatterning microscope (3-s exposure) after focusing on the glass/buffer interface. The flow cell was subsequently washed with three flow cell volumes of carbonate buffer, and fibrinogen-biotin was then injected at 20 µg/ml in 0.1 M sodium bicarbonate, pH 8.3. After a 2-min incubation, the chamber was washed with three flow cell volumes of carbonate buffer, and NeutrAvidin (or NeutrAvidin-Dylight-550) was injected at 50 µg/ml in 0.1 M sodium bicarbonate, pH 8.3. The chamber was then washed with Hepes buffer (10 mM Hepes, pH 7.6) and quenched with PLL-PEG (0.2 mg/ml in 10 mM Hepes, pH 7.6) for 2 min. The chamber was then washed with Hepes buffer, and BSA-biotin-Alexa647 was added (10 µg/ml in 10 mM Hepes, pH 7.6) for 1 min, before extensive washing with Hepes buffer before imaging. For controls where NeutrAvidin was directly micropatterned, fibrinogen-biotin was replaced by NeutrAvidin/NeutrAvidin-DyLite550 (50 µg/ml in carbonate buffer), and the chamber was directly washed with Hepes buffer, quenched with PLL-PEG, and incubated with BSA-biotin-Alexa647 as above.

In general, we found that buffer composition (Fig. S1 B) and avoiding drying of the surface (Fig. S4, A and B) were fundamental for ensuring reproducibility, selectivity, and homogeneity of micropatterning. Note that while the micropatterning

buffer is important, the use of fibrinogen anchors allows one to bind the proteins of interest to the micropattern in virtually any buffer, as this binding step occurs after the micropatterning process.

## Image processing

Images were processed using Fiji (Schindelin et al., 2012). Figures were assembled in Adobe Illustrator 2019.

Patterning selectivity and homogeneity were computed as follows per micropattern:

$$Selectivity = \frac{AvgIntensity_{pattern} - Avgbackground_{camera}}{AvgIntensity_{notpattern} - Avgbackground_{camera}}$$

$$Homogeneity = \frac{AvgIntensity_{pattern} - Avgbackground_{camera}}{\sqrt{varIntensity_{pattern} - varbackground_{camera}}},$$

with $AvgIntensity_{pattern}$ representing the average fluorescence intensity of the fluorescent protein onto a micropatterned region of interest (ROI) and its associated variance, $varIntensity_{pattern}$; $AvgIntensity_{notpattern}$ representing the average fluorescence intensity of the fluorescent protein onto a nonpatterned ROI adjacent to the micropattern; and $Avgbackground_{camera}$ representing the average background intensity of the camera and its associated variance, $varbackground_{camera}$. $AvgIntensity_{pattern}$, $AvgIntensity_{notpattern}$, and $varIntensity_{pattern}$ were measured in an ROI of identical size (and as large as possible to provide good estimates). $Avgbackground_{camera}$ and $varbackground_{camera}$ were obtained by measuring the signal in the dark upon screwing a lid onto the camera. We verified that PLL-PEG does not display any autofluorescence signal in the conditions we used (in other words, the fluorescence intensity of an unpatterned PLL-PEG coverslip that was never incubated with any fluorescent protein is virtually identical to the camera background).

Similarly, we measured a proxy of the amount of protein specifically being deposited onto the micropatterned ROI as follows:

$$Amount\ of\ protein\ patterned = AvgIntensity_{pattern} - AvgIntensity_{notpattern}$$

Note that while the *Selectivity* is a normalized value and can be compared between fluorophores, the *Amount of protein patterned* is only valid when comparing the same fluorophore, as it will also depend on the photophysics of the dye and the sensitivity of the instrument to the dye.

To evaluate the amount of cross-adsorption between micropatterns during sequential micropatterning of multiple fibrinogens (Fig. 1, C–E; and Fig. S1, D–F), we thought to express the amounts of fibrinogens going to the *wrong* micropatterns as a fraction of the amount going to the *right* micropattern. This allows direct comparison between fibrinogens and gives a direct idea of how much cross-contamination there is. For instance, in Fig. 1, C–E, we sequentially micropatterned fibrinogen-ATTO488, fibrinogen-Alexa546, and then fibrinogen-Alexa647. We thus measured the fluorescence of each fibrinogen in all three micropatterns and normalized these values with respect to the intended micropattern (i.e., since the order is [1] fibrinogen-ATTO488, [2] fibrinogen-Alexa546, then [3] fibrinogen-Alexa647, signals in the ATTO488 channel are normalized to micropattern 1,

Alexa546 to micropattern 2, and Alexa647 to micropattern 3). In mathematical terms,

$$Normalized\ patterned\ Fibrinogen\ ATTO488_{pattern\ \#i} =$$
$$\frac{AvgIntensity^{488}_{pattern\ \#i} - Avgbackground^{488}_{coverslip}}{AvgIntensity^{488}_{pattern\ \#1} - Avgbackground^{488}_{coverslip}}$$
$$Normalized\ patterned\ Fibrinogen\ Alexa546_{pattern\ \#i} =$$
$$\frac{AvgIntensity^{546}_{pattern\ \#i} - Avgbackground^{546}_{coverslip}}{AvgIntensity^{546}_{pattern\ \#2} - Avgbackground^{546}_{coverslip}}$$
$$Normalized\ patterned\ Fibrinogen\ Alexa647_{pattern\ \#i} =$$
$$\frac{AvgIntensity^{647}_{pattern\ \#i} - Avgbackground^{647}_{coverslip}}{AvgIntensity^{647}_{pattern\ \#3} - Avgbackground^{647}_{coverslip}},$$

with $AvgIntensity^{488}_{pattern\ \#i}$ (respectively, $AvgIntensity^{546}_{pattern\ \#i}$ and $AvgIntensity^{647}_{pattern\ \#i}$) the average fluorescence signal in the ATTO488 channel in the micropatterned ROI (respectively, in the Alexa546 and Alexa647 channels) in the micropattern #i, and $Avgbackground^{488}_{coverslip}$ (respectively, $Avgbackground^{546}_{coverslip}$ and $Avgbackground^{647}_{coverslip}$) the average fluorescence background in the ATTO488 channel (respectively, in the Alexa546 and Alexa647 channels) measured on a PLL-PEG–coated coverslip never exposed to UV or incubated with fluorescent fibrinogens. As noted before, due to the absence of autofluorescence, this is virtually identical to the background gray levels of the camera ($Avgbackground_{camera} \approx Avgbackground^{488\ /\ 546\ /647}_{coverslip}$). As can be seen in Fig. 1 E, cross-adsorption between the different micropatterns is minimal: the amount of fibrinogen-Alexa546 wrongly going onto the fibrinogen-ATTO488 micropattern is only 1.3 ± 0.05% (mean ± SEM; n = 12) of that of the amount of fibrinogen-Alexa546 going to the intended, fibrinogen-Alexa546, micropattern. Similarly, the amount of fibrinogen-Alexa647 going to the wrong fibrinogen-ATTO488 micropattern (respectively, fibrinogen-Alexa546) is only 0.7 ± 0.04% (respectively, 3.0 ± 0.3%; n = 12) of the fibrinogen-Alexa647 going to the right fibrinogen-Alexa647 micropattern. Similar results were obtained in sequential micropatterning of two (not three) fibrinogens (see Fig. S1 F, where the graph scale has been split to better appreciate the difference).

To evaluate the amount of fibrinogen binding nonspecifically to unexposed PLL-PEG (Fig. S1 F), we computed the amounts of fibrinogens found on the PLL-PEG coverslip as a fraction of the amount going to the intended micropatterned region. Specifically, in the fibrinogen-ATTO488/Alexa647 sequential micropatterning experiment described above, we also measured the average ATTO488 and Alexa647 signals in ROIs corresponding to unpatterned regions. Importantly, we maintained the position of the ROIs in the field of view (that is, we moved the sample to an area without micropatterns) in order to account for any potential inhomogeneities in the illumination of our microscope. Keeping the above nomenclature, we thus evaluated

$$Normalized\ Fibrinogen\ ATTO488_{PLL-PEG} =$$
$$\frac{AvgIntensity^{488}_{PLL-PEG} - Avgbackground^{488}_{coverslip}}{AvgIntensity^{488}_{pattern\ \#1} - Avgbackground^{488}_{coverslip}}$$

$$Normalized\ Fibrinogen\ Alexa647_{PLL-PEG} =$$

$$\frac{AvgIntensity_{PLL-PEG}^{647} - Avgbackground_{coverslip}^{647}}{AvgIntensity_{pattern\ \#2}^{647} - Avgbackground_{coverslip}^{647}},$$

with $AvgIntensity_{PLL-PEG}^{488}$ the average fluorescence signal in the ATTO488 channel in the same ROI used to measure $AvgIntensity_{pattern\ \#1}^{488}$ but in a region of the sample where there is no micropattern (same thing for $AvgIntensity_{PLL-PEG}^{647}$ and $AvgIntensity_{pattern\ \#2}^{647}$). As can be seen in Fig. S1 F, nonspecific binding to the PLL-PEG coverslip is extremely low: the amount of fibrinogen-ATTO488 wrongly going onto the nonmicropatterned PLL-PEG is only 0.40 ± 0.06% ($n = 8$) that of the amount of fibrinogen-ATTO488 going to its intended micropattern (respectively, 0.66 ± 0.04% for fibrinogen-Alexa647; $n = 8$). This is in good agreement with the fact that when the image dynamic range is adjusted to see the minute signals on the PLL-PEG part, a punctate, single molecule–like signal is observed (arrowheads in Fig. S1 E, right panel).

The enrichment of mScarlet relocalization (Fig. 6 C) was quantified as follows:

$$mScarlet\ enrichment\ in\ inner\ pattern =$$

$$\frac{AvgIntensity_{mScarlet\ in\ GFP\ pattern\ ROI} - Avgbackground_{non\ pattern}}{AvgIntensity_{mScarlet\ not\ in\ GFP\ pattern\ ROI} - Avgbackground_{non\ pattern}},$$

with $AvgIntensity_{mScarlet\ in\ GFP\ pattern\ ROI}$ representing the average fluorescence intensity of mScarlet in the ROI corresponding to the micropatterned GFP, and $AvgIntensity_{mScarlet\ not\ in\ GFP\ pattern\ ROI}$ representing the average fluorescence intensity of mScarlet in the ROI of the same size but corresponding to a ROI in the fibronectin micropattern surrounding the GFP center. $Avgbackground_{non\ pattern}$ represents the mScarlet fluorescence intensity in unpatterned regions of the coverslip.

The enrichment of EGFR and clathrin (Fig. 7 C) were calculated as follows:

$$Enrichment\ in\ center\ pattern =$$

$$\frac{AvgIntensity_{center\ pattern\ ROI} - AvgIntensity_{cell-free\ center\ pattern\ ROI}}{AvgIntensity_{surrounding\ ROI} - AvgIntensity_{cell-free\ surrounding\ ROI}},$$

with $AvgIntensity_{center\ pattern\ ROI}$ representing the average fluorescence intensity of EGFR/clathrin in the ROI corresponding to the micropatterned EGF (where a cell is adhered to the micropattern), $AvgIntensity_{cell-free\ center\ pattern\ ROI}$ representing the fluorescence signal in a neighboring ROI corresponding to micropatterned ligand (but without a cell adhered, to account for potential fluorescence bleed-through into the imaging channel), $AvgIntensity_{surrounding\ ROI}$ representing the EGFR/clathrin signal in the fibronectin micropattern surrounding the EGF center (where a cell is adhered), and $AvgIntensity_{cell-free\ surrounding\ ROI}$ representing the fluorescence intensity in a neighboring fibronectin micropattern without a cell adhered (again, to account for any potential fluorescence bleed-through).

The fold enrichment of GFP-Notch1 (Fig. 7 F) was calculated from live imaging of U2OS cells landing and spread on micropatterns, 20 min after landing on micropatterns, as follows:

$$Enrichment\ in\ center\ pattern =$$

$$\frac{AvgIntensity_{center\ pattern\ ROI\ 20\ minutes} - AvgIntensity_{center\ pattern\ ROI\ 0\ minute}}{AvgIntensity_{surrounding\ ROI\ 20\ minutes} - AvgIntensity_{surrounding\ ROI\ 0\ minute}},$$

where $AvgIntensity_{center\ pattern\ ROI\ 20\ minutes}$ is the average fluorescence intensity of GFP-Notch1 20 min after a cell landing on the micropattern in the region corresponding to the micropatterned DLL4, and $AvgIntensity_{center\ pattern\ ROI\ 0\ minutes}$ represents the average fluorescence intensity in the same region before the cell landing on the micropattern (to account for any potential fluorescence bleed-through). Similarly, $AvgIntensity_{surrounding\ ROI\ 20\ minutes}$ is the average fluorescence intensity of GFP-Notch1 in the outer micropattern, 20 min after a cell lands on the micropattern, and $AvgIntensity_{surrounding\ ROI\ 0\ minutes}$ is the average fluorescence intensity of the same region before a cell landing on it.

To quantify the recruitment of GFP-Notch1 to DLL4 micropatterns over time (Fig. S5 E), the raw increase in GFP-Notch1 signal overlying the central, DLL4 region compared with the outer fibronectin micropattern was calculated (due to relatively low signals, this is less volatile than the fold change):

$$Increase\ of\ GFP - Notch\ signal\ in\ center\ pattern =$$
$$(AvgIntensity_{center\ pattern\ ROI} - AvgIntensity_{center\ pattern\ ROI\ 0\ minute}) -$$
$$(AvgIntensity_{surrounding\ ROI} - AvgIntensity_{surrounding\ ROI\ 0\ minute}),$$

where $AvgIntensity_{center\ pattern\ ROI}$ is the average fluorescence intensity of GFP-Notch1 in the central, DLL4 micropattern at the specific time point, and $AvgIntensity_{center\ pattern\ ROI\ 0\ minutes}$ represents the average fluorescence intensity in the central region before a cell landing (to account for any potential fluorescence bleed-through). $AvgIntensity_{surrounding\ ROI}$ is the average fluorescence intensity of GFP-Notch1 in the outer micropattern at the same time point, and $AvgIntensity_{surrounding\ ROI\ 0\ minutes}$ is the average fluorescence intensity in the same region before a cell landing on the micropattern (to account for potential bleed-through).

For analysis of MT gliding on Kin1-biotin micropatterns (Fig. 4), movies were projected in time, and these projections were used to define the path of gliding MTs, from which kymographs were generated and analyzed using the "Kymo Toolbox" plugin for ImageJ, developed by Fabrice Cordelières (Zala et al., 2013). For analysis of the proportion of MTs undergoing directional movement, MTs were manually segmented based on whether or not they showed clear, directed motion.

To quantify the increased micropatterning efficiency of S2 cells onto small micropatterns in Fig. 5, we computed the normalized density of micropatterned cells as follows:

$$Normalized\ patterned\ cell\ density =$$

$$\frac{nCell_{pattern}/Area_{pattern}}{mean(nCell_{pattern}^{ConA}/Area_{pattern}^{ConA})},$$

with $nCell_{pattern}$ representing the number of cells per micropattern and $Area_{pattern}$ representing the area of the same micropatterns, and $mean(nCell_{pattern}^{ConA}/Area_{pattern}^{ConA})$ the mean density of the Con A sample. Only small micropatterns, of diameter 20–40 μm, were considered for analysis.

To quantify the specificity of S2 cell adhesion, we computed the normalized nonspecific adhesion (to micropatterns of between 20 and 40 µm in diameter) as follows:

$$Normalized\ nonspecific\ adhesion = \left(1 - \frac{nCell_{pattern}/Area_{pattern}}{nCell_{nonpattern}/Area_{nonpattern} + nCell_{pattern}/Area_{pattern}}\right) \times 100.$$

### Statistics

Unless stated otherwise, measurements are given in mean ± SEM. Statistical analyses were performed using GraphPad Prism 8 or SigmaStat 3.5 with an α of 0.05. Normality of variables was verified with Kolmogorov–Smirnov tests. Homoscedasticity of variables was always verified when conducting parametric tests. Post hoc tests are indicated in their respective figure legends.

### Online supplemental material

Fig. S1 shows fibrinogen micropatterning using LIMAP. Fig. S2 illustrates that fibrinogen is readily functionalized with target proteins or ligands. Fig. S3 shows that fibrinogen anchors improve selectivity and homogeneity of micropatterns on PEG-silane surfaces. Fig. S4 displays the effect of sample drying on micropatterning efficiency and improved micropatterning of S2 cells by fibrinogen anchors. Fig. S5 illustrates the dynamics and controls of subcellular micropatterning of receptors. Video 1 shows that fibrinogen anchors facilitate micropatterning of active motors. Video 2 demonstrates the dynamics of GBP-TM-mScarlet relocalization by fibrinogen-GFP micropatterns. Video 3 illustrates the dynamics of GFP-Notch relocalization by fibrinogen-Delta micropatterns.

## Acknowledgments

We thank Vicente Jose Planelles-Herrero for help with graphics, and Nicolas Chiaruttini, Jerome Boulanger, James Manton, Christopher Rowlands, and Clemens Kaminski for sharing their expertise in optics and microscope design. We thank Manuel Théry and Benoit Vianay for their extensive help in setting up micropatterning in the laboratory at the beginning of this project. We thank Laurent Blanchoin and Antoine Jegou for critically reading the manuscript. We thank the electronics workshop of the Laboratory of Molecular Biology, in particular Martin Kyte, for building custom LED drivers for this project. We thank the technical instrumentation workshop of the Laboratory of Molecular Biology, in particular Steve Scotcher and Adam Fowle, for their help in designing and manufacturing the custom-made optical mounts of the DMD illuminator. We thank Tino Pleiner and Mark Allen for plasmids.

This work has been supported by the Medical Research Council (MC_UP_1201/13 to E. Derivery) and the Human Frontier Science Program (Career Development Award CDA00034/2017-C to E. Derivery). J.L. Watson is the recipient of a Michael Neuberger Studentship from the Max Perutz Fund and Trinity College, Cambridge.

S.C. Blacklow receives research funding for an unrelated project from Novartis, is a member of the scientific advisory board of Erasca, Inc., is an advisor to MPM Capital, and is a consultant on unrelated projects for IFM, Scorpion Therapeutics, and Ayala Therapeutics. The other authors declare no competing financial interests.

Author contributions: J.L. Watson performed all micropatterning experiments except Fig. S3, which was performed by S. Aich. J.L. Watson purified all proteins and fibrinogen anchors, with some assistance from E. Derivery. E. Derivery developed the LED illuminator with assistance from J.L. Watson and S. Aich. B. Oller-Salvia, with support from J. Chin, developed the one-step quantitative synthesis of the BBTB. A.A. Drabek and S.C. Blacklow provided the purified DLL4 protein and derived the GFP-Notch1 U2OS cell line. J.L. Watson, S. Aich, and E. Derivery produced the figures. E. Derivery wrote the manuscript. All authors discussed the results and commented on the manuscript.

Submitted: 15 September 2020

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

**Figure S1. Fibrinogen micropatterning using LIMAP. (A)** Optical design of the DMD-UV illuminator used in this study. Schematic optical path. A 385-nm high-power UV LED light source is collimated using an AR-coated aspheric lens, and the collimated UV beam is then directed toward a DMD chip at a 24° angle of incidence (corresponding to twice the tilting angle of the DMD mirrors). The image of the DMD chip is then relayed onto the conjugate of the sample plane at the backport of the microscope through a 4f imaging system (f1 = f2 = 125 mm UV fused silica bi-convex lenses, AR-coated). This intermediate image is then relayed onto the sample plane by a tube lens and the objective. To combine DMD-UV illumination with TIRF illumination, a 470-nm dichroic is placed after f2 within a custom backport assembly (Cairn). To offer a second illumination wavelength for 450-nm optogenetic stimulation, our design also contains a second 450-nm collimated LED at the symmetric −24° angle. This LED can be exchanged for any other LED to provide epifluorescence imaging. **(B and C)** Enhanced fibrinogen micropatterning efficiency depends on the buffer used. **(B)** PLL-PEG–coated glass was processed for LIMAP patterning with identical UV exposure, micropattern shape, and photoinitiator concentration (50 mM BBTB in 0.1 M sodium bicarbonate, pH 8.3). Fibrinogen-Alexa546 (50 µg/ml) was then adsorbed onto the UV-activated surface in either PBS or carbonate buffer. After washing, red fluorescence of the patterns was imaged by TIRFM using identical settings. **(C)** Quantification of the effects seen in B (average selectivity ± SEM; see Materials and methods). Fibrinogen quantitatively patterns better in carbonate buffer. Statistics were performed using a Mann–Whitney rank-sum test. n, number of patterns measured. **(D–F)** Advantages of fibrinogen for multiplexed micropatterning. **(D)** Scheme illustrating the different steps for sequential multiplexed micropatterning of two fibrinogens labeled with different fluorophores (ATTO488 and Alexa647). **(E)** Multiplexed micropatterning of fibrinogen-ATTO488 and Alexa647 (50 µg/ml) using the scheme depicted in D onto PLL-PEG–coated glass. Note that there is high specificity of the fibrinogen for their specific pattern and minimum overlap among the two fluorescent fibrinogens. In addition, binding of fibrinogen to unexposed PLL-PEG is minimal, down to a punctate, single molecule–like level (orange arrowheads). **(F)** Quantification of the effects seen in E: the fluorescence of each Fibrinogen was measured on the two patterns and normalized to the fluorescence of their intended pattern (i.e., first pattern for Alexa647 and second for ATTO488; mean ± SEM). Statistics were performed using a Kruskal–Wallis test followed by a Dunn post hoc test (P < 0.0001). n, number of patterns measured. Note that the vertical scale of the graph is split to better appreciate the minute amounts of fibrinogen deposited onto the nonintended patterns or the unpatterned PLL-PEG area. Scale bars, 10 µm. n.s., not significant.

## A  General three-step synthesis : Fibrinogen GFP / Fibrinogen NeutrAvidin / Fibrinogen ConA

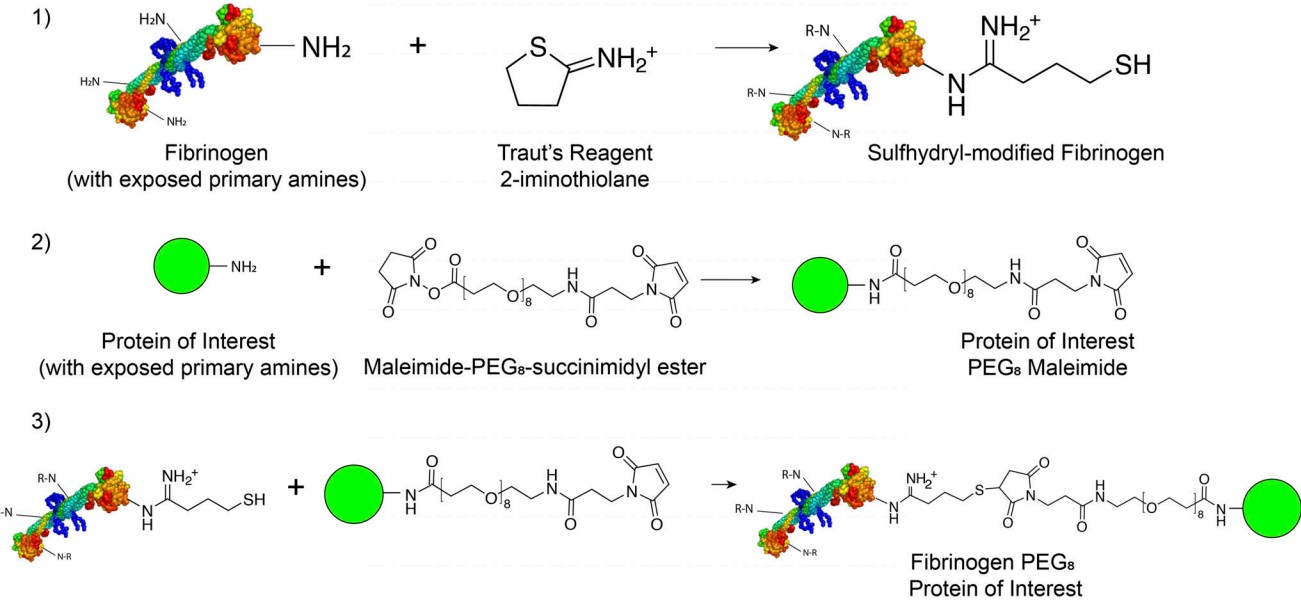

## B  Two-step synthesis for proteins/coumponds with exposed thiols : Fibrinogen GBP

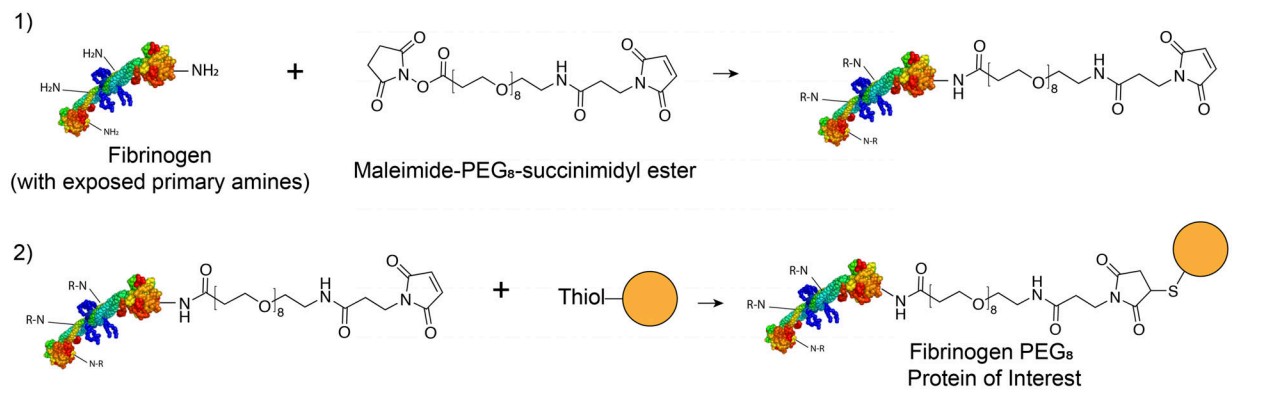

## C  One-step synthesis : : Fibrinogen biotin

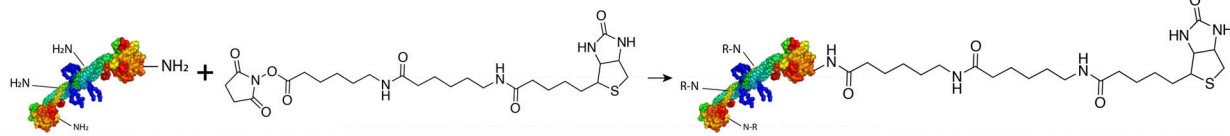

Figure S2. **Fibrinogen is readily functionalized with target proteins or ligands. (A)** General three-step method to functionalize fibrinogen with a protein of interest using lysine reactivity. Reactive thiols are generated onto fibrinogen, which are then reacted with the maleimide-functionalized protein of interest (GFP, Con A, NeutrAvidin). Excess unreacted protein of interest is removed by differential native precipitation. **(B)** Two-step method to functionalize fibrinogen if the protein/ligand of interest has reactive thiols (for example, nanobodies engineered with cysteines). **(C)** One-step method to functionalize fibrinogen with an NHS derivative of a ligand of interest (biotin, fluorophores).

**Figure S3.** **Fibrinogen anchors improves selectivity and homogeneity of micropatterns on PEG-silane surfaces. (A)** NeutrAvidin-DyLight-550 (50 µg/ml) was micropatterned on PLL-PEG–coated glass using LIMAP. Alternatively, fibrinogen-biotin was micropatterned with identical UV exposure, pattern shape, and protein concentration, followed by the addition of NeutrAvidin-Dylight-550 and imaging by TIRFM. **(B)** Quantification of amount of micropatterned protein (left), pattern selectivity (middle), and homogeneity (right) in the sample presented in A (mean ± SEM). Statistics were performed using a Mann–Whitney test for the left and right panels, and a Student's *t* test for the middle panel. Fibrinogen-biotin significantly improves the micropatterning of NeutrAvidin-Dylight-550. **(C)** NeutrAvidin, NeutrAvidin-Dylight-550, or fibrinogen-biotin were micropatterned on PEG-silane–coated glass using LIMAP with identical UV exposure, pattern shape, and protein concentration (50 µg/ml). After pattern quenching and washing, BSA-biotin-Alexa647 (5 µg/ml) was added for 5 min. The sample was then washed, and BSA-biotin-Alexa647 fluorescence was imaged by TIRFM (NeutrAvidin, NeutrAvidin-Dylight-550, and fibrinogen-biotin samples). Alternatively, fibrinogen-biotin was micropatterned as above, and NeutrAvidin (or NeutrAvidin-Dylight-550) was added before addition of BSA-biotin-Alexa647 and TIRFM imaging. Two different exposures were used for each sample (top versus bottom line) so that each lane could be represented with the same dynamic range. **(D)** Quantification of the amount of micropatterned protein (left), micropattern selectivity (middle), and homogeneity (right) in the sample presented in C (mean ± SEM). Statistics were performed using a one-way ANOVA test followed by a Tukey post hoc test after log10 transformation of the data (P < 0.001). Fibrinogen-biotin enhances significantly the amount of BSA-biotin-Alexa647 patterned, as well as pattern selectivity and homogeneity. While fibrinogen-biotin efficiently improves the micropatterning of NeutrAvidin-Dylight-550 compared with direct micropatterning, this does not translate into improved micropatterning of BSA-biotin-Alexa647, likely because NeutrAvidin-Dylight-550 has lost some biotin-binding activity due to its fluorescent labeling. Scale bar, 10 µm. biot, biotin; n.s., not significant.

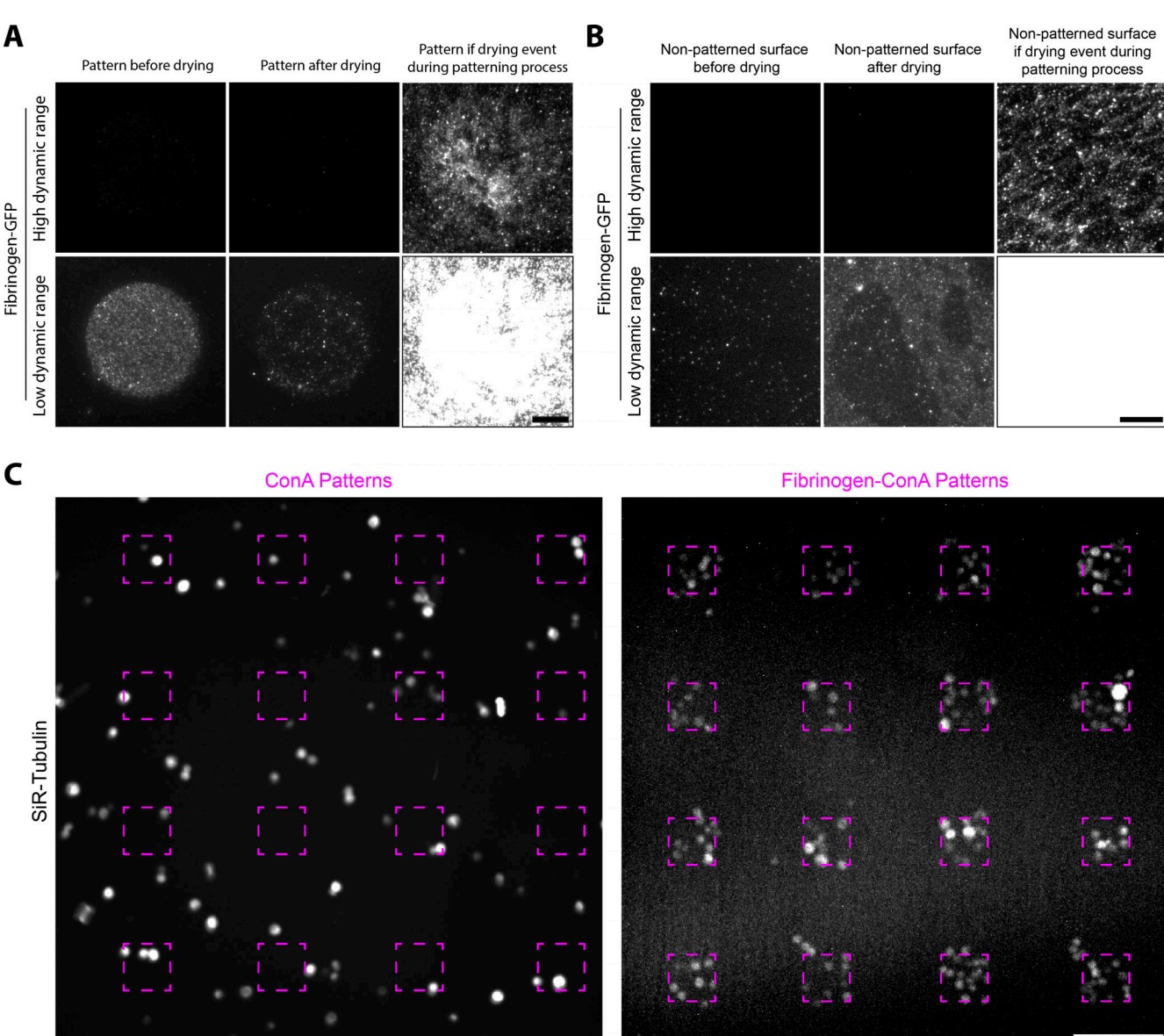

Figure S4. **Effect of sample drying on micropatterning efficiency and improved micropatterning of S2 cells by fibrinogen anchors. (A and B)** Sample drying during or after the micropatterning process affects micropatterning efficiency. **(A)** Drying of the pattern during or after the micropatterning process negatively impacts the micropatterning efficiency. Left: PLL-PEG–coated glass was processed for LIMAP patterning of fibrinogen-GFP (50 µg/ml). Middle: After imaging the pattern, the sample was dried, then rehydrated and imaged again in the same conditions. Right: PLL-PEG–coated glass was processed for LIMAP patterning of fibrinogen-GFP as before, but the sample was allowed to dry after the adsorption process. Two different dynamic ranges for visualization were used for each sample (top versus bottom line) so that each lane could be represented with the same dynamic range. **(B)** Protein adsorption to the non-micropatterned region is also increased upon drying during/after the micropatterning process. Sample was treated as in A, but nonpatterned regions were imaged. **(C)** Fibrinogen–Con A enhances the micropatterning of S2 cells. Con A (doped with 10% rhodamine–Con A) or fibrinogen–Con A (doped with 10% fibrinogen–Alexa546) was micropatterned at 50 µg/ml onto PLL-PEG–coated glass using deep UV and a chromium mask. Coverslips were washed, and S2 cells were added for 1 h before addition of SiR-tubulin, to label cells, for 30 min. After washing, cells and micropatterns were imaged by spinning-disc confocal microscopy. Micropatterning efficiency of S2 cells is lower on Con A compared with fibrinogen–Con A. Dashed line, region exposed to deep UV. Scale bars, 10 µm (A and B), and 100 µm (C).

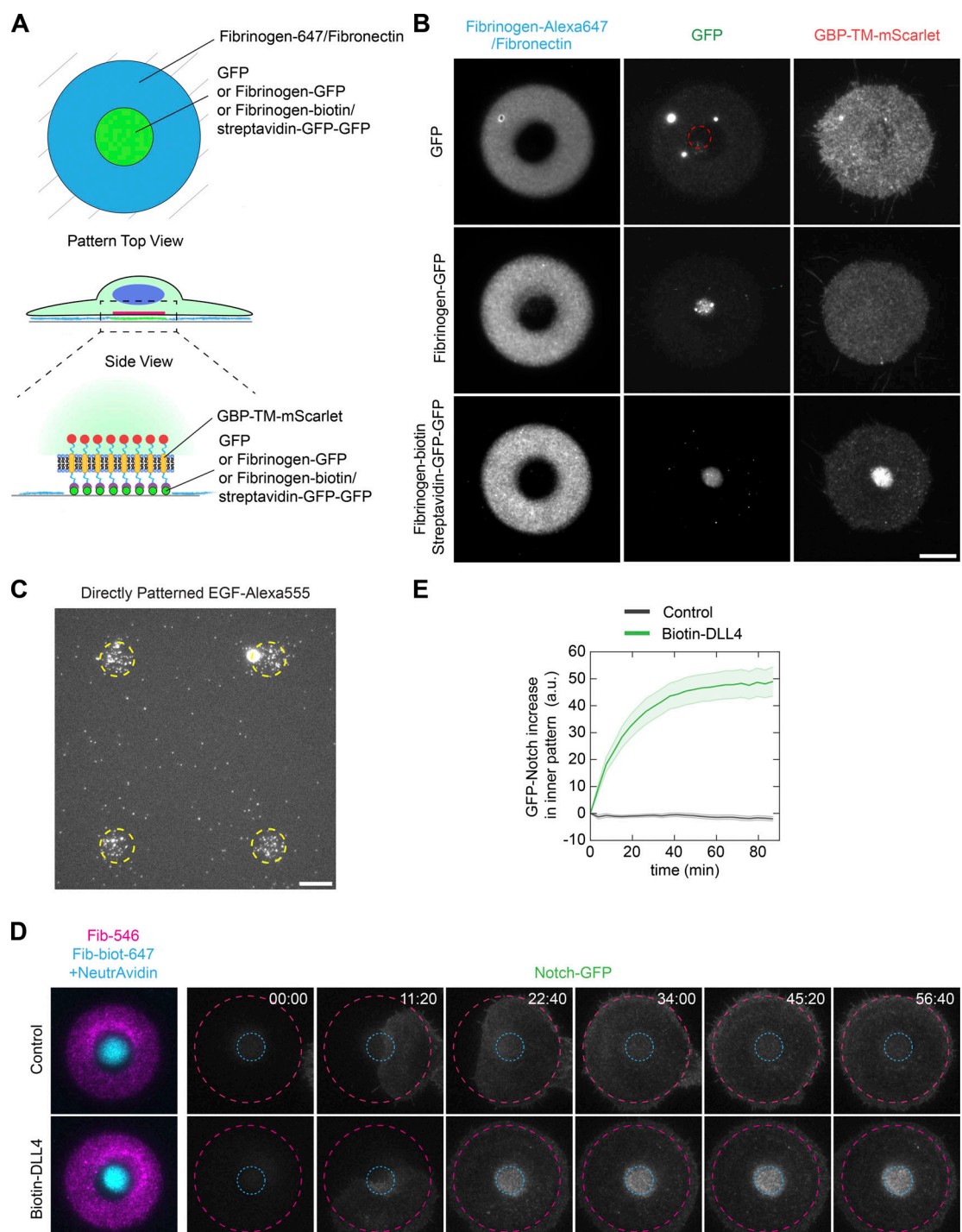

Figure S5. **Dynamics and controls of subcellular micropatterning of receptors. (A–C)** Regular micropatterning does not allow subcellular micropatterning of receptors. **(A)** Experimental scheme. Stable NIH/3T3 cells constitutively expressing GBP-TM-mScarlet were allowed to spread on dual patterns of fibronectin/fibrinogen-Alexa647 and either GFP, fibrinogen-GFP (low degree of labeling of 0.5 mol GFP per mol fibrinogen), or fibrinogen-biotin::streptavidin-GFP-GFP, then imaged live by TIRF microscopy. **(B)** Only high-density GFP micropatterning via fibrinogen-biotin::streptavidin-GFP-GFP allows efficient relocalization of the GBP-TM-mScarlet construct onto an area defined by the extracellular pattern. Note that bottom panel corresponds to Fig. 6 B, reproduced here for convenience. Note also that the dynamic range of the GFP channel panels is not identical here. There is much more GFP when using streptavidin-GFP-GFP compared with using fibrinogen-GFP. **(C)** In contrast to when biotin-EGF was attached to the fibrinogen-biotin-ATTO490LS::NeutrAvidin sandwich (Fig. 7, A–C), direct micropatterning of biotin-EGF::streptavidin-Alexa555 (1 µg/ml) showed very weak, inhomogeneous, and nonspecific micropatterning, preventing micropatterning of the second, surrounding fibronectin pattern. Dashed line, region exposed to UV. **(D and E)** Dynamics of GFP-Notch relocalization by Delta micropatterns. **(D)** U2OS cells stably expressing GFP-Notch1 were allowed to spread on dual patterns of fibronectin/fibrinogen-Alexa647 and fibrinogen-biotin-ATTO490LS::NeutrAvidin::biotin-DLL4 and GFP-Notch fluorescence was imaged live during spreading by TIRF microscopy. Elapsed time in minutes:seconds. **(E)** Quantification of the effects seen in D (mean ± SEM; number of cells analyzed: 27 for control and 28 for Biotin-DLL4); see also Materials and methods. Scale bars, 10 µm. Fib, fibrinogen; biot, biotin.

Video 1. **Fibrinogen anchors facilitate micropatterning of active motors.** Biotinylated-Kinesin1 motors (Kin1-Biotin) were micropatterned on PLL-PEG–coated glass using LIMAP either directly or indirectly through a fibrinogen-biotin-ATTO490LS::NeutrAvidin sandwich. After washing and quenching, GMPCPP-stabilized fluorescent MTs were added in the presence of ATP and their motion observed by TIRFM. Dashed purple line delineates the kinesin pattern as imaged either through post-labeling of Kin1-Biotin by streptavidin-Alexa647 for direct micropatterning or by fibrinogen-biotin-ATTO490LS fluorescence for the indirect labeling. This movie corresponds to Fig. 4. Scale bar, 10 µm. sec, seconds. Playback speed: 30 frames per second.

Video 2. **Dynamics of GBP-TM-mScarlet relocalization by fibrinogen-GFP micropatterns.** NIH/3T3 cells stably expressing GBP-TM-mScarlet were allowed to spread on dual micropatterns of fibronectin/fibrinogen-Alexa647 and fibrinogen-biotin-ATTO490LS::streptavidin-GFP-GFP and were then imaged live by TIRF microscopy. Elapsed time in minutes. This movie corresponds to Fig. 6 D. Scale bar, 10 µm. Playback speed: 10 frames per second.

Video 3. **Dynamics of GFP-Notch relocalization by fibrinogen-Delta micropatterns.** U2OS cells stably expressing GFP-Notch1 were allowed to spread on dual micropatterns of fibronectin/fibrinogen-Alexa647 and fibrinogen-biotin-ATTO490LS::NeutrAvidin::biotin-DLL4 (see Fig. 7, D and E), and GFP-Notch fluorescence was imaged live during spreading by TIRF microscopy. Elapsed time in minutes. This movie corresponds to Fig. S5 D. Scale bar, 10 µm. Playback speed: 10 frames per second.

