## [Peer Review File · The Journal of Cell Biology]

High efficacy subcellular micropatterning of proteins using fibrinogen anchors

Joseph Watson, Samya Aich, Benjami Oller-Salvia, Andrew Drabek, Stephen Blacklow, Jason Chin, and Emmanuel Derivery

Corresponding Author(s): Emmanuel Derivery, MRC Laboratory of Molecular Biology

Review Timeline:

Submission Date:	2020-09-15
Editorial Decision:	2020-10-29
Revision Received:	2020-11-16
Accepted:	2020-11-23

Monitoring Editor: Kenneth Yamada

Scientific Editor: Dan Simon

Transaction Report:

DOI: <https://doi.org/10.1083/jcb.202009063>

October 29, 2020

RE: JCB Manuscript #202009063

Dr. Emmanuel Derivery
MRC Laboratory of Molecular Biology
Cell Biology
Francis Crick Avenue
Cambridge CB2 0QH
United Kingdom

Dear Dr. Derivery,

Thank you for submitting your manuscript entitled "Fibrinogen anchors for micropatterning of active proteins and subcellular receptor relocalisation" to the Journal of Cell Biology. The manuscript has now been assessed by two expert leaders in the field, whose reports are appended below.

As you can see, the reviewers are enthusiastic about this Tools manuscript and recommend only minor revisions. We would thus be happy to publish your paper in JCB pending final revisions necessary to meet our formatting guidelines (see details below).

Please also address the specific concerns of these expert reviewers, including the questions about cross-contamination and clarifications, as well as consider addressing the questions about affinities and image adjustment. In addition, please comment on the point about intrinsic biological activity of fibrinogen, which I also wondered about at our preliminary Editorial evaluation stage.

A. MANUSCRIPT ORGANIZATION AND FORMATTING:

Full guidelines are available on our Instructions for Authors page, <https://jcb.rupress.org/submission-guidelines#revised>. **Submission of a paper that does not conform to JCB guidelines will delay the acceptance of your manuscript.**

- 1) Text limits: Character count for Articles and Tools is < 40,000, not including spaces. Count includes title page, abstract, introduction, results, discussion, and acknowledgments. Count does not include materials and methods, figure legends, references, tables, or supplemental legends.
- 2) Figures limits: Articles and Tools may have up to 10 main text figures.
- 3) Figure formatting: Scale bars must be present on all microscopy images, including inset magnifications. Molecular weight or nucleic acid size markers must be included on all gel electrophoresis.
- 4) Statistical analysis: Error bars on graphic representations of numerical data must be clearly described in the figure legend. The number of independent data points (n) represented in a graph

must be indicated in the legend. Statistical methods should be explained in full in the materials and methods. For figures presenting pooled data the statistical measure should be defined in the figure legends. Please also be sure to indicate the statistical tests used in each of your experiments (both in the figure legend itself and in a separate methods section) as well as the parameters of the test (for example, if you ran a t-test, please indicate if it was one- or two-sided, etc.). Also, if you used parametric tests, please indicate if the data distribution was tested for normality (and if so, how). If not, you must state something to the effect that "Data distribution was assumed to be normal but this was not formally tested."

5) Materials and methods: Should be comprehensive and not simply reference a previous publication for details on how an experiment was performed. Please provide full descriptions (at least in brief) in the text for readers who may not have access to referenced manuscripts. The text should not refer to methods "...as previously described."

6) Please be sure to provide the sequences for all of your primers/oligos and RNAi constructs in the materials and methods. You must also indicate in the methods the source, species, and catalog numbers (where appropriate) for all of your antibodies.

7) Microscope image acquisition: The following information must be provided about the acquisition and processing of images:

- a. Make and model of microscope
- b. Type, magnification, and numerical aperture of the objective lenses
- c. Temperature
- d. Imaging medium
- e. Fluorochromes
- f. Camera make and model
- g. Acquisition software
- h. Any software used for image processing subsequent to data acquisition. Please include details and types of operations involved (e.g., type of deconvolution, 3D reconstitutions, surface or volume rendering, gamma adjustments, etc.).

8) References: There is no limit to the number of references cited in a manuscript. References should be cited parenthetically in the text by author and year of publication. Abbreviate the names of journals according to PubMed.

9) Supplemental materials: There are strict limits on the allowable amount of supplemental data. Articles/Tools may have up to 5 supplemental figures. Please also note that tables, like figures, should be provided as individual, editable files. A summary of all supplemental material should appear at the end of the Materials and methods section. Your manuscript currently exceed this limit but, in this case, we will be able to give you the extra space but please try not to add to the current total.

10) eTOC summary: A ~40-50 word summary that describes the context and significance of the findings for a general readership should be included on the title page. The statement should be written in the present tense and refer to the work in the third person. It should begin with "First author name(s) et al..." to match our preferred style.

11) Conflict of interest statement: JCB requires inclusion of a statement in the acknowledgements regarding competing financial interests. If no competing financial interests exist, please include the following statement: "The authors declare no competing financial interests." If competing interests

are declared, please follow your statement of these competing interests with the following statement: "The authors declare no further competing financial interests."

12) A separate author contribution section is required following the Acknowledgments in all research manuscripts. All authors should be mentioned and designated by their first and middle initials and full surnames. We encourage use of the CRediT nomenclature (<https://casrai.org/credit/>).

13) ORCID IDs: ORCID IDs are unique identifiers allowing researchers to create a record of their various scholarly contributions in a single place. At resubmission of your final files, please consider providing an ORCID ID for as many contributing authors as possible.

B. FINAL FILES:

-- High-resolution figure and video files: See our detailed guidelines for preparing your production-ready images, <https://jcb.rupress.org/fig-vid-guidelines>.

Thank you for this interesting contribution, we look forward to publishing your paper in Journal of Cell Biology.

Sincerely,

Kenneth Yamada, MD, PhD

Editor
Journal of Cell Biology

Dan Simon, PhD
Scientific Editor
Journal of Cell Biology

Reviewer #1 (Comments to the Authors (Required)):

In this article, Watson and colleagues present a superb methodological work. This work will bring the technique of micropatterning to a new level, making it broadly accessible to biology labs and giving it a much broader application range than it currently has. The authors quantitatively analyse the output of their method and demonstrate very original and new applications. The method is based on a combination of classical micropatterning techniques and clever molecular biology, to produce a variety of molecular anchors with both specificity and high patterning quality. I thus strongly recommend publication of this article. It does not need any revision to be published and will be highly valuable for the cell biology community.

One point the authors might want to add, is a discussion of the binding strength of the various anchors they propose, as this might be also exploited to produce patterns which can be detached or not depending on the force exerted by the cell. This is also a potential limitation if it is not well controlled (as mentioned in the cited article by Fink et al, in which strong enough cells are able to rip off patterned molecules if they are not anchored strongly enough). Another interesting development would be to combine such molecular anchors with force sensors, either DNA or FRET based (which could then be genetically encoded as well).

Reviewer #2 (Comments to the Authors (Required)):

The manuscript submitted by Watson et al. describes how micropatterning of proteins can be made more efficient. Direct coating of proteins to surfaces can be difficult because not every protein sticks equally well to surfaces, which causes problems particularly when coating small areas (as it is the case for micropatterning). Here, authors show that such problems can be circumvented by the use of fibrinogen. The advantages are manifold:

- i) Fibrinogen is sticky and binds homogeneously to glass surfaces
- ii) Fibrinogen is cheap and can be purchased in large quantities.
- iii) It can be functionalized to selectively bind other components
- iv) Functionalization with different tags allows efficient micropatterning of two (or even more) different components.
- v) Using fibrinogen first as anchor for other proteins can preserve their bioactivity.

Experiments supporting the various points are well chosen, well described and quantified. To my opinion, the presented method provides an effective toolkit that enables setting up experiments to answer in a relatively easy and reproducible manner a number of important cell biological questions. The manuscript is therefore of great interest to the readership of JCB and suitable for publication after addressing the following minor points.

1) Figure 2 describes the sequential coating of functionalized fibrinogen but it is not entirely clear how much cross-contamination there is. One can assume that the second fibrinogen coating (Fibrinogen-Neutravidin) can also bind to some extent the first microprinted area (coated with Fibrinogen-GBP). If I understood it correctly, some of this has been quantified (Figure 2c and d) but I find it difficult to understand precisely how. From the figures it seems that selectivity of BSA-BiotinA647 is much higher than the selectivity of GFP to its ligand?

A helpful experiment to analyse cross-contamination of the two fibrinogens may be direct labelling of fibrinogen with two dyes and then compare how much of the second coated fibrinogen binds to the area with the first one.

2) It would be good to know how sticky fibrinogen is to the PLL-PEG area. Fig 4a shows a relatively strong binding to the PLL-PEG coated area (Fig 4a)?

3) One of the potential drawbacks of using fibrinogen is that it itself is a "bioactive" component. Cells can bind to it, can cleave it and thus its use may have limitations. This should be discussed in the relevant section.

4) Some of the figures (e.g. Fig 2b) would be better visible with adjusted brightness/contrast enhancement.

Dear Dr. Simon,

Please find enclosed our revised manuscript.

We were very pleased to read the general enthusiasm of the reviewers for our work. We are grateful to both of them for raising a number of very interesting points to add to the discussion of our manuscript. In particular, the potential activity of Fibrinogen itself, which is also something you mentioned in your letter, is obviously something very important to discuss for the community that we had overlooked in our first version (just because it did not interfere in our experiments it does not mean it will *never* interfere!). We also conducted the excellent experiment suggested by Reviewer #2 to directly measure cross-contamination between Fibrinogen species during multiplexed patterning, which we generalized to three proteins. Similarly, we experimentally measured Fibrinogen “stickiness” to PLL-PEG substrates. These two new experiments further highlighted the beneficial properties of Fibrinogen that make it a good micropatterning anchor, and as such are a great addition to the paper. These new experiments constitute new panels in Figures 1, 3 and S1, as well as text modifications to emphasize these key findings (highlighted in yellow in the text). We also reformatted the manuscript and figures to fit your guidelines and added to the “acknowledgments” section of our manuscript the disclosure of financial interests of one co-author for unrelated projects.

In conclusion, we think that these additional data and discussion nicely highlight why using Fibrinogen as an anchor is useful for functional micropatterning of cells, and as such will improve the impact of our manuscript. Following is a detailed response to the comments of the referees.

Reviewer #1

In this article, Watson and colleagues present a superb methodological work. This work will bring the technique of micropatterning to a new level, making it broadly accessible to biology labs and giving it a much broader application range than it currently has. The authors quantitatively analyse the output of their method and demonstrate very original and new applications. The method is based on a combination of classical micropatterning techniques and clever molecular biology, to produce a variety of molecular anchors with both specificity and high patterning quality. I thus strongly recommend publication of this article. It does not need any revision to be published and will be highly valuable for the cell biology community.

We thank this reviewer for this very positive assessment of our work!

One point the authors might want to add, is a discussion of the binding strength of the various anchors they propose, as this might be also exploited to produce patterns which can be detached or not depending on the force exerted by the cell. this is also a potential limitation if it is not well controlled (as mentioned in the cited article by Fink et al, in which strong enough cells are able to rip off patterned molecules if they are not anchored strongly enough). Another interesting development would be to combine such molecular anchors with force sensors, either DNA or FRET based (which could then be genetically encoded as well).

We thank this reviewer for pointing out these excellent discussion matters, (see also our reply to point 2.3, as reviewer #2 has a partially overlapping comment).

When considering the binding strength of our Fibrinogen anchors, one must consider at least two things: 1) the anchoring of the fibrinogen to the glass substrate, 2) anchoring of the cell to the anchor (that is EGFR binding to EGF, or, alternatively the cell binding directly to fibrinogen itself).

The nice thing about always using Fibrinogen is that the first component of binding (aka adsorption to the micropattern) is probably virtually identical between anchors. On the other hand, the strength of the second component will vary according to the specific binding pair chosen: some antibody/antigen interactions might be stronger than some ligand/receptor for instance. As all these remain non-covalent interactions, the reviewer is completely right that when the cell is pulling strongly on an anchor, if the binding strength of the cell for the anchor is stronger than the anchor for the glass, then cells could rip off the micropatterns, as previously observed for Fibronectin in the work of Fink and colleagues (Fink et al. 2007).

Importantly, we never observed such “ripping off” in the experiments presented in this paper, which suggest that in the timeframe of our experiments (1-2hrs) the forces generated by adhesion and endocytosis in HeLa, 3T3 and U2OS cells are weaker than the binding-strength of the Fibrinogen anchors for the glass. But obviously this does not mean that this will always be the case for other cells, or for longer experiments, which we agree needs to be discussed in the paper. We envision that reticulating the micropatterns using mild fixative may help in cases where the cell rips off the patterns. Conversely, as suggested by this reviewer, one could tune down strength of binding of the anchor to the cell (using weaker nanobodies for instance) to engineer patterns that would unbind from the cell when it pulls too much.

Coupling force sensors to Fibrinogen anchors is a fantastic idea, and we are grateful for this reviewer to have shared it with us. In particular, we think that it would be amazing to couple this to our asymmetric micropatterns (Figs. 6-7). There are indeed a few ways of doing this, either by coupling purified proteins modified to the FRET force sensors (LaCroix et al., 2018) or using DNA origamis (Blakely et al., 2014). Along the same lines, one could also think about coupling Fibrinogen to force inducers that are controlled by light, as described in a recent preprint (Zheng et al., 2020). Importantly, all these mechanosensing/inducing molecules can be readily coupled to Fibrinogen using the protocols we established in this paper. Obviously, for this to work, one would need to ensure that the glass/anchor component of the binding is as strong as it can be (at least stronger than the binding of the cell to the anchor), and, similarly one would have to make sure that the cell does not bind to Fibrinogen itself (using S2 cells for instance).

A new paragraph has been added to the discussion to reflect these important points for the community, reprinted here for convenience.

While not a concern for in vitro experiments, Fibrinogen might not always be the anchor of choice when working with cells, as it is itself a bioactive molecule. First, obviously, Fibrinogen anchors are probably not suited to study cell types involved in the biology of Fibrinogen, like platelets. Indeed, any secreted thrombin activity would likely render the anchors non-functional. Furthermore, while some cells do not bind to Fibrinogen, such as S2 cells (see Fig.5), Fibrinogen has been widely used to functionalize surfaces to allow cell adhesion. If cells just bind to Fibrinogen as they would bind to extracellular matrix proteins (like Fibronectin), then using Fibrinogen as an anchor to micropattern ligands asymmetrically (like we do in Figs. 6-7) would actually be an advantage, as the density of adhesion sites could be kept identical between the Fibrinogen and Fibrinogen-ligand part of the micropatterns. However, if one wants to investigate cell mechanics by functionalizing Fibrinogen with force sensors (LaCroix et al., 2018; Blakely et al., 2014), or light-controlled force inducers (Zheng et al.,

2020), then the direct binding of the cell to the Fibrinogen itself rather than the force-sensing/inducing moiety would be an issue, as it will interfere with the experiments. Another potential concern for these experiments is that if cells are able to bind to Fibrinogen (or Fibrinogen anchors), then strong enough cells might be able to rip off the micropatterns over time, as was shown previously with Fibronectin (Fink et al., 2007). Indeed, as good as Fibrinogen micropatterns are, they are still achieved by protein adsorption. While we did not observe this in any of the experiments described in this paper, as all Fibrinogen micropatterns presented in this study were stable when interacting with 3T3, HeLa and U2OS cells (Figs. 6-7), this might be a concern for researchers working with other cells and/or much longer experiments. A solution to this potential problem could be to use mild aldehyde or NHS fixatives to cross-link the Fibrinogen molecules of the micropattern prior to adding cells. Indeed, Fibrinogen functionalized in the conditions described here indeed still offer reactive amines, a property we exploited to generate bi-functional Fibrinogens like Fibrinogen-Biotin-ATTO490LS. Conversely, one could also tune down the interaction between the anchor and the cell, using weaker nanobodies for instance, so the anchor would unbind when the cell pulls too much on it. These potential caveats notwithstanding, now that we have identified the key properties that make a good micropatterning anchor, it would be an interesting avenue of future research to find an alternative to Fibrinogen that has no biological activity.

Reviewer #2

The manuscript submitted by Watson et al. describes how micropatterning of proteins can be made more efficient. Direct coating of proteins to surfaces can be difficult because not every protein sticks equally well to surfaces., which causes problems particularly when coating small areas (as it is the case for micropatterning). Here, authors show that such problems can be circumvented by the use of fibrinogen. The advantages are manifold:

- i) Fibrinogen is sticky and binds homogeneously to glass surfaces
- ii) Fibrinogen is cheap and can be purchased in large quantities.
- iii) It can be functionalized to selectively bind other components
- iv) Functionalization with different tags allows efficient micropatterning of two (or even more) different components.
- v) Using fibrinogen first as anchor for other proteins can preserve their bioactivity.

Experiments supporting the various points are well chosen, well described and quantified. To my opinion, the presented method provides an effective toolkit that enables setting up experiments to answer in a relatively easy and reproducible manner a number of important cell biological questions. The manuscript is therefore of great interest to the readership of JCB and suitable for publication after addressing the following minor points.

We thank this reviewer for this nice and very positive overview of our work.

2.1) Figure 2 describes the sequential coating of functionalized fibrinogen but it is not entirely clear how much cross-contamination there is. One can assume that the second fibrinogen coating (Fibrinogen-Neutravidin) can also bind to some extent the first microprinted area (coated with Fibrinogen-GBP). If I understood it correctly, some of this has been quantified (Figure 2c and d) but I find it difficult to understand precisely how. From the figures it seems that selectivity of BSA-BiotinA647 is much higher than the selectivity of GFP to its ligand?

A helpful experiment to analyse cross-contamination of the two fibrinogens may be direct labelling of fibrinogen with two dyes and then compare how much of the second coated fibrinogen binds to the area with the first one.

We thank the reviewer for this excellent comment and key experimental suggestion.

Indeed, the reviewer is perfectly right that in our previous Figure 2 (now Figure 3), the cross-contamination we measured was actually the convolution of two components: first, the binding of each Fibrinogen anchor to the wrong pattern, and second, the binding of each target protein to the wrong anchor, or to the wrong pattern if improperly quenched. Even if in the end what matters for the experiments is the cross contamination of the target proteins, and not that of the anchors, it was an imprecise way to measure the low cross-contamination between different Fibrinogens and thus did not properly demonstrate why Fibrinogen was such a good candidate for an anchor. What is key is to demonstrate here that first, Fibrinogen is not “sticky” for itself when micropatterned (this is a bit counter-intuitive, as Fibrinogen is known for its tendency to aggregate), and second, Fibrinogen patterns can be efficiently quenched. Indeed, the combination of both properties ensuring reliable sequential multiplexed micropatterning. While we had indirectly inferred these two properties in our previous version, we agree with the reviewer that their direct demonstration was very important.

We thus performed the experiment suggested by this reviewer, meaning labelling Fibrinogen with different dyes and sequentially micropatterning them. We performed this experiment with two Fibrinogens (new Fig. S1D-E), but also generalised this to three Fibrinogens (new Fig.1C-D) to further highlight these key properties of Fibrinogen. As can be seen in these figures, the vast majority of each Fibrinogen “flavour” ended up in its intended micropattern with minimal overlap onto the other patterns.

To quantify this cross-contamination in a more intuitive way, which was also a concern of this reviewer, we thought to express the amounts of Fibrinogens going to the *wrong* patterns as a fraction of the amount going to the *right* pattern (meaning that for instance the normalized Fibrinogen-ATTO488 fluorescence is 1 on its intended micropattern, and hopefully less than this on the others). This allows direct comparison between Fibrinogens, and gives a direct idea of how much cross-contamination there is. As can be seen in Fig.1E, cross adsorption between the different patterns is minimal: the amount of Fibrinogen-Alexa546 wrongly going onto the Fibrinogen-ATTO488 pattern is only $1.3 \pm 0.05\%$ ($n=12$) of the amount going to its intended pattern. Similarly, the amount of Fibrinogen-Alexa647 wrongly going to the Fibrinogen-ATTO488 pattern is only $0.7 \pm 0.04\%$, and the amount wrongly going onto the Fibrinogen-Alexa546 is only $3.0 \pm 0.3\%$ ($n=12$). Similar results were obtained in sequential patterning of two (not three) Fibrinogens (Fig. S1F, where the graph scale has been split to better appreciate the difference).

We also adopted the same intuitive quantification to evaluate the (convolved) cross-contamination between proteins of interest micropatterned using Fibrinogen Anchors (new Figure 3C). This confirmed the minute levels of cross-contamination between the two proteins: the amount of GFP wrongly going onto the Biotin-BSA-Alexa647 micropattern is only $5.0 \pm 0.3\%$ ($n=7$) of the amount going to its intended micropattern. Similarly, the amount of Biotin-BSA-Alexa647 going to the wrong GFP pattern is only $0.9 \pm 0.1\%$ ($n=7$) of the amount going to its intended micropattern.

This new set of key experiments nicely highlight two properties of fibrinogen making it a protein of choice for a multiplexed micropatterning: first, micropatterned Fibrinogen really does not stick to Fibrinogen in solution, and second, Fibrinogen micropatterns can be very efficiently quenched to

exhibit only very minor cross-contamination. We modified the main text and the methods to showcase these key new results.

2.2) It would be good to know how sticky fibrinogen is to the PLL-PEG area. Fig 4a shows a relatively strong binding to the PLL-PEG coated area (Fig 4a)?

We thank the reviewer for this excellent point. Indeed, when we quantified patterning efficiency throughout our study, we measured the ratio between the amount of protein on the pattern over the amount of protein on the PLL-PEG. But obviously by doing this normalisation, we lose the information of how much is actually bound onto the PLL-PEG. So, to further make the point that Fibrinogen is a good candidate to make an anchor, we agree with this reviewer that evaluating its “stickiness” to PLL-PEG is a very important point.

We built on the experiment suggested by this reviewer in point **2.1** and did these measurements for two fluorescent Fibrinogens within a multiplexed micropatterning experiment (new Fig. S1D). As before, we expressed the amount of Fibrinogen binding to PLL-PEG areas of the coverslip as a fraction of the amount in its intended micropatterned region. To measure these low signals as accurately as possible, we took particular care in this experiment to measure the instrument background by measuring the autofluorescence of a PLL-PEG coated coverslip having never been exposed to UV nor incubated with fluorescent Fibrinogen imaged in the same conditions. Similarly, Fibrinogen fluorescence on PLL-PEG areas was measured in Regions Of Interest (ROI) at the exact same position of the microscope’s field of view in order to take into consideration any potential inhomogeneities in the illumination profile of our instrument.

As can be seen in Fig. S1D, non-specific binding to the PLL-PEG coverslip is near background levels: the amount of Fibrinogen-ATTO488 wrongly going onto the non-micropatterned PLL-PEG is only $0.40 \pm 0.06\%$ ($n=8$) of the amount of Fibrinogen-ATTO488 going to its intended pattern (respectively $0.66 \pm 0.04\%$, for Fibrinogen-Alexa647, $n=8$). This is in good agreement with the fact that when the image dynamic range is adjusted to see the low signals on the PLL-PEG part, a punctate, single-molecule-like signal is observed (arrowheads in new Fig. S1C, right panel).

This demonstrates that we are not in a situation where Fibrinogen binds everywhere, although better on micropatterned regions, but rather that Fibrinogen binds nearly exclusively onto the intended micropatterned region, which is obviously a key advantage for *in vivo* and *in vitro* work. It is worth noting that these minute levels of Fibrinogen binding to PLL-PEG do not reach a density that is enough to “do” anything. For instance, S2 cells do not adhere to the PLL-PEG regions in the Fibrinogen-ConA patterning experiment (now in Fig. 5), and Microtubules do not glide on the PLL-PEG region in the Fibrinogen-biotin::neutravidin::biotin-kinesin experiment (now in Fig. 4). Altogether, these new data highlight a key additional property of Fibrinogen that makes it a good anchor: its negligible binding to PLL-PEG.

Independently of this important result, we realized that what might also have spurred this comment was a potential confusion in the specific panel referred here (former Fig. 4A, now Fig. 5A). In this panel, the intensity on the “pattern” channel was never meant to be compared between samples. But it is now obvious that the arbitrary dynamic range we had chosen for each image could be confusing within the context of the whole multipanel montage (see also our reply to point **2.4**). We have now corrected this in the revised version, as well as put a disclaimer in the legend these images that these images should not be compared between themselves. Note that this does not affect the conclusions of this experiment in any way, as what we quantified here is the number of cells per pattern, not the absolute

fluorescence of either: providing we use the adequate dynamic range, both the patterns and the cells can easily be identified in all conditions. We do apologize for this potential confusion.

2.3) One of the potential drawbacks of using fibrinogen is that it itself is a "bioactive" component. Cells can bind to it, can cleave it and thus its use may have limitations. This should be discussed in the relevant section.

We thank this reviewer for this excellent point, which was also raised by the editor.

As foreseen by this reviewer, we also think that the pros or cons of using Fibrinogen as an anchor in cell biology experiments would actually vary depending on whether cells simply *adhere* to it, *pull* on it (meaning can exert force on it), or *act* on it (meaning they can degrade it).

On the one hand, while not all cells bind to Fibrinogen (S2 cells, for instance, don't, see Fig.5), Fibrinogen has indeed been widely used to functionalize surfaces to allow cell adhesion. Importantly, if cells just bind to Fibrinogen as they would bind to other extracellular matrix proteins (like fibronectin), then using Fibrinogen as an anchor to micropattern ligands asymmetrically (like we do in Figures 6-7) would actually be an advantage, as the density of adhesion sites for the cell could be kept identical between the Fibrinogen and Fibrinogen-ligand part of the patterns. The main problem we envision Fibrinogen-binding would be if one wants to measure forces, as functionalizing Fibrinogen with some kind of force-sensing or force-inducing moiety would obviously not work if cells directly bind to Fibrinogen and the force sensor/inducer (see our answer to Reviewer #1).

On the other hand, if cells bind to Fibrinogen, there is also the possibility that strong-enough cells could rip off the Fibrinogen micropattern from the glass, as previously observed for Fibronectin (Fink et al. 2007; see also our reply to Reviewer #1). While this was not a problem with the cells we used (HeLa, 3T3 and U2OS), as all our micropatterns were stable for more than the time frame of the experiment, this may be a problem with other cells or longer experiments. We envision that reticulation of the Fibrinogen micropattern using mild fixatives could help improve micropattern stability in this case.

Finally, it is completely right that Fibrinogen anchors probably cannot be used with cells involved in Fibrinogen biology (like platelets). Indeed, if cells secrete some kind of Fibrinogen modifying enzymes, like thrombin, it is likely that Fibrinogen anchors will be compromised.

These concerns notwithstanding, now that we have identified the key properties required for a given protein to make a good micropatterning anchor, it would be an interesting avenue of future research to find an alternative to Fibrinogen that has no biological activity, to alleviate these issues. A new paragraph has been added to the discussion to reflect these important points for the community (this paragraph has been added to our reply to Reviewer #1 for Reviewers' convenience).

2.4) Some of the figures (e.g. Fig 2b) would be better visible with adjusted brightness/contrast enhancement.

Throughout the paper, we decided to display two versions of key panels: a low dynamic range to see how little background/cross-contamination signal there is, and a high dynamic range panel to appreciate quantitatively how well proteins micropattern. We do agree with the reviewer that to highlight the low level of cross-contamination between Fibrinogens, which was the point of Fig. 2B

(now 3B), having a duplicated version of the panels where the dynamic range is reduced would help the reader appreciate how little the amount of protein on the “wrong” pattern is.

However, since cross-contamination is now much better quantified (new Figs. 1D-E, 3C and S1E-F), and since we did add a set of low-dynamic range images in Fig. S1E to visually appreciate this specific point, we feel that adding a low-dynamic range version of all six panels Fig. 3B would render this figure more confusing for the reader and would rather keep it as it is. This being said, we took care in our revised version to adjust the brightness/contrast of the displayed panels for the reader to improve the visibility of our images (see our reply to point **2.2**).